# Pasture-Based Livestock Economics under Joint Production of Commodities and Private Amenity Self-Consumption: Testing in Large Nonindustrial Privately Owned *Dehesa* Case Studies in Andalusia, Spain

**Pablo Campos \*, Bruno Mesa and Alejandro Álvarez**

Spanish National Research Council (CSIC), Institute for Public Goods and Policies (IPP), C/Albasanz, 26–28, E-28037 Madrid, Spain; bruno.mesa@cchs.csic.es (B.M.); alejandro.alvarez@cchs.csic.es (A.Á.)
\* Correspondence: pablo.campos@csic.es; Tel.: +34-91-602-2535

**Abstract:** In this study, we apply the hypothesis of private amenity which simulates that the nonindustrial livestock farmers are assured an ex-ante normal minimum operating profitability rate for their investments in the production systems of livestock species based on grazing in a case study of *dehesas* in Andalusia, Spain. The ex-post measurement in the Agroforestry Accounting System of the commercial operating opportunity cost incurred by the owners at the close of the period corresponds to the lower limit of the additional amount of noncommercial intermediate product of the private amenity self-consumption service (ISSnca). When the livestock farmers obtain an above-normal operating profitability rate, it is assumed that the absence of opportunity cost results in the free use of the private amenity and, therefore, the latter is a free (noneconomic) service with zero value. In the case study of *dehesa* farms, the results show that the commercial operating profitability rates at basic prices are below the normal. When the ISSnca is included, the operating profitability rates at social prices for the livestock species exceed by 30%, on average, the assumed normal rate of 3%. However, due to the decline in the prices of the inanimate fixed capital in 2010, the average total profitability rate for the livestock species is estimated at 0.1%, which differs substantially from the assumed normal operating profitability rate. These results are of interest with regard to the design and application of official economic accounts at farm scale, which, as in the European Commission Farm Accounting Data Network, omit the measurement of ISSnca.

**Keywords:** Farm Accounting Data Network; government compensations; Agroforestry Accounting System; net value added at social price; ordinary cash flow

## 1. Introduction

The European Union policies for rural development highlight the preservation of extensive livestock activity in areas of high natural value with the aim of mitigating the loss of unique domestic biological variety threatened with extinction, while also favouring economic activity in rural villages [1]. The planning of budget cycles in the European Union periodically leads to arguments concerning government policies for economic investment in the livestock farming activity, with these debates often being most heated in the negotiations for compensations under the Common Agricultural Policy (CAP).

The individual nonindustrial owners of large private *dehesa* farms in Andalusia demand compensation from the government for the additional production of public goods and services of the agro-silvo-pastoral landscape, which provides continuity for livestock grazing in extensive livestock farm production [2]. This demand from the livestock farmers is based on the hypothesis that the commercial products of extensive livestock farming are not competitive, and, to mitigate the decline in livestock grazing demanded by society,

the livestock farmers should receive fair compensation from the government for the contribution of the livestock farming to the increased offer of public goods and services in working landscapes which are consumed without direct payment by the beneficiaries [3]. In this economic context of public and private products of extensive livestock farming, governments are faced with the choice between accepting the tendency toward future decline in the supply of public products if livestock grazing continues to diminish or mitigating/avoiding the decline by increasing compensations.

The main novelty with regard to extending the total product in the Agroforestry Accounting System (AAS) application to the case study of *dehesas* is the incorporation of government compensations (ISSncc) and the private amenities (ISSnca) as noncommercial intermediate products of services (ISSnc). The incorporation of the concept of ISSnca, first defined in [4], based on the voluntary opportunity cost of the livestock farm owners, is the innovation which most affects the economic results for individual livestock species in situations where the normal net operating margins exceed the respective commercial residuals, estimated by their basic prices.

The official Farm Accounting Data Network (FADN) methodology gives the values added, at basic prices, of the commercial goods of the agricultural and livestock farms during the period, aggregated according to the main type of product of the farms [5]. In other words, for those in which extensive livestock farming predominates, the commercial values added in the FADN are not presented separately for each livestock species or livestock farming activity.

Hence, the academic literature on extensive livestock farming generally presents the aggregate value added at basic prices for the economic activities of the farms in which livestock farming predominates, without incorporating modifications into the official FADN methodology [6,7]. Other authors have presented qualitative reviews of the literature on livestock rearing in extensive systems without including economic results [3,8]. The exception to these omissions in the literature can be found in our publications which apply the AAS to farms with a predominance of tree species of the *Quercus* [4,9,10] and *Pinus* [11] genera.

The aim of the application of the AAS methodology to the case study livestock species at the real management scale of an individual *dehesa* farm in 2010 was to illustrate, through economic results captured in the field, the rationales behind the livestock investments of nonindustrial individual private owners of large farms. The purpose of the case studies is to further our microeconomic understanding of the qualitative interpretations of the incentives of the owners of large private *dehesas* where agricultural land and pasture only make up a small part of the total farm area. These *dehesas* mainly comprise holm oak (*Quercus ilex* L.) and cork oak (*Quercus suber* L.) open woodland, along with small amounts of other tree species (e.g., wild olive trees, riparian forest, and eucalyptus).

The objective of this research was to test the modelling of the economic rationales for the livestock species production systems based on natural grazing in large private *dehesa* farms belonging to nonindustrial owners of the land and livestock in Andalusia, Spain. This objective required the incorporation of noncommercial intermediate products of private amenity services (ISSnca) [4]. These services are based on the economic rationale of the livestock farmers, characterized by prior voluntary acceptance of the possibility of receiving a commercial net operating margin at a basic price below the normal amount received in exchange for guaranteed personal and family self-consumption of the private amenities provided by the livestock species which graze in their *dehesa* farms. The nonindustrial owners immobilize their livestock investments with the aim of joint production of commercial goods and self-consumption of private amenity services, with productive links with other activities in the *dehesa*. Hence, the challenge in this research was to validate the hypothesis that the immobilized investment in the rearing of a livestock species, based on grazing of the farmer's own land, fulfils the *dehesa* farmer's expectation of attaining at least the normal net operating margin at the close of the period.

The organization of this research continues as follows: Section 2 includes a summary of the *dehesa* landscapes of Andalusia along with a brief description of the main characteristics of the case study farms and the livestock farming activity which takes place on them. Section 2 compiles and describes the main economic indicators estimated. In Section 3, we present and comment on the results obtained in this research. In Section 4, we discuss the implications for government policy implementation. Lastly, Section 5 presents the main conclusions of this research.

## 2. Materials and Methods

The materials and methods applied in the large nonindustrial privately owned *dehesa* case studies in Andalusia, Spain are presented in this section. First, we define open woodlands, known as *dehesas*, where the farm case studies are located. Then, we briefly describe the livestock species studied and the main characteristics of the livestock management in the *dehesa* case studies (Section 2.1).

Second, we present the most relevant accounting framework of the Agoforestry Accounting System (AAS) applied in the large nonindustrial privately owned *dehesa* case studies (Section 2.2). This should help readers' comprehension of the text without a need to turn to previously published literature. The main variables described here are the following: (i) total product; (ii) noncommercial intermediate product of private amenity self-consumption; (iii) intermediate consumption; (iv) forage unit livestock consumption; (v) total income; (vi) net value added; (vii) capital gain; (viii) net operating margin; (ix) profitability rates; (x) ordinary cash flow. In addition, we compare the net value added under the Farm Accounting Data Network (FADN) and the AAS.

### 2.1. Case Studies of Pasture-Based Livestock Farming on Large Nonindustrial Privately Owned Dehesas in Andalusia, Spain

The livestock species reared on natural pasture are those which have led to the formation of the open woodland working landscapes comprising trees of the *Quercus* genus (*Quercus ilex* L., *Qurecus suber* L., *Quercus faginea* Lam., *Quercus pyrenaica* Willd.) along with small areas of other species (e.g., *Olea europaea* L., *Fraxinus angustifolia* Vahl.) in the five regions in the west, center, and south of Spain (Figure A1, Appendix A). These working landscapes account for 61% of the total area of 3,606,154 ha of the 112,000 agro-silvo-pastoral farms known as *dehesas* in Spain ([4] Table 2, p. 3). In Andalusia, the open woodland working landscapes of the 4408 *dehesas* occupy 62% of the total area of 743,775 ha. The 1009 *dehesas* of more than 200 ha in Andalusia contain 63% of the open woodland working landscapes and 68% of the total *dehesa* area ([12] Table 23, p. 46).

In this study, we applied the Agroforestry Accounting System (AAS) separately to the bovine, ovine, caprine, porcine, equine, and apicultural livestock species that graze on the large nonindustrial privately owned *dehesa* case-study farms in Andalusia, Spain. The open woodland working landscapes make up 77% of the total area of 15,372 ha of these *dehesas* case studies, with open woodlands of holm oak and cork oak occupying 46% and 20%, respectively, of the total *dehesa* case-study area. Due to the relatively small area covered by the case-study *dehesas*, it is not possible to transfer the statistically significant results obtained for the livestock activity to the total area of the large *dehesas* of Andalusia. Nevertheless, the results for the case-study *dehesas* do provide a qualitative illustration of the economic rationale trends present in the extensive livestock-rearing activity of the large silvo-pastoral farms of Andalusia belonging to individual private nonindustrial owners.

Livestock grazing occurs in all the area occupied by the *Quercus* genus on the farms and it may also take place to a far smaller extent in 4% of the forested area of the *Pinus* genus on the case-study *dehesas*. The grazed fodder consumed by the livestock species comprises mainly grasses, twigs, and wild fruits (e.g., acorns). In several of the case-study *dehesas*, Iberian pigs are still fattened by feeding on acorns (termed *montanera*). Acorns

account for 11.9% of the 288.5 FU/ha grazed forage units (FUs) consumed by the livestock in the period (Table A1, Appendix A).

The livestock species reared on the *dehesas* are native or crosses with other foreign breeds of meat cattle and fighting bulls, meat sheep, meat and milk goats, Iberian fattened and suckling pigs of the breed "*Negro Entrepelado*", horses for breeding, and bees. The studied farms usually rear several species, with the most common being meat cattle. With regard to bovine, ovine, and caprine species, livestock production is mainly directed toward the sale of weaned offspring to be fattened and slaughtered away from the farm, as well as the rearing of fighting bulls, in cycles of 2–5 years required in the legislation, for subsequent use in festivals and bullfights. The production of porcine species is directed toward final fattening of Iberian pigs or *montanera* and a token amount of extensive breeding of suckling pigs of the breed "*Negro Entrepelado*". With regard to the equine species, the pure-bred Andalusian, *Hispano-Arabe*, and Anglo-Arabic breeds are sold as studs for breeding and recreational use by the owner, with mules and donkeys being used for pulling and other tasks. The production of honey and beeswax is destined for sale and a small amount for consumption by the bees.

The labor in the livestock activity of the *dehesas* is mainly paid labor (employees). The exception is the labor associated with the goats and beekeeping activity, which is mainly self-employed, family labor (Table A2, Appendix A).

The concept of livestock species stocking rates applied to the definition of livestock rearing according to the consumption of natural pasture is ambiguous, since pasture forage may be supplemented by trough feeding. This is the case of existing livestock rearing in the private *dehesas* studied, where an average of 57.3% of the metabolizable energy of their food in 2010 came from supplements, such that, in the case-study *dehesas*, stabled fattening of livestock is not practiced and only a small number of breeding sows of *montanera* pigs do not graze. The large area occupied by woodland, exceeding 80% of the total area, the halting of grass growth in the summer, the *montanera* or fattening of pigs on holm oak acorns in the autumn, and the predominance of bovine species both for meat and bullfighting result in a livestock stocking rate which on average reaches 0.44 livestock units (LU) per hectare, which is notably lower than the upper limit of 1.4 LU/ha considered in the literature to define extensive livestock rearing ([3] p. 1365).

Apart from the livestock activity, other activities are undertaken in the studied *dehesas* which give the owners the greatest profitability margins. These activities include cork production, grazing, private amenity, and hunting. It should be noted that the game species in the case-study *dehesas* compete with the domestic livestock in terms of grazing, consuming similar amounts of pasture (Table A1, Appendix A). The main game species on these farms are deer and wild boar, although there are other species such as the Iberian ibex, mouflon, fallow deer, roe deer, partridge, rabbit, and other small game migrant species such as the thrush or pigeon.

*2.2. Economics of Private Livestock Farming under the Hypothesis of Amenity Self-Consumption*

In this section, we present the concepts of economic rationales of private nonindustrial owners for investment in extensive rearing of livestock species, along with the estimation of total income for these species. We focus on the case-study *dehesas* in terms of internal economic exchanges of noncommercial intermediate product of private amenity self-consumption services (ISSnca) and government compensations (ISSncc) with the final products consumed of private amenity (FPcaa) and public landscape (FPcla) valued through the Agroforestry Accounting System (AAS). We summarize the similarities and differences in the values added between the AAS and the official Farm Accounting Data Network (FADN) methodologies.

2.2.1. Total Product Factorial Allocations

The exclusivity of the livestock owner over the ownership of the products and the transmission of property rights to third parties are the initial conditions for identifying and estimating the exchange value of the eligible products (with and without market prices) of the livestock activity, which give the total income. The total product (TP) is composed of the intermediate product (IP) and the final product (FP). The final product is made up of the final product consumed (FPc) and the own-account gross capital formation (GCF) of the livestock activity. The total product consumed (TPc) is obtained by adding the IP and the FPc.

The TPc incorporates the ordinary total cost (TCo) of the total cost (TC) of the livestock activity, while the total cost of investment (TCi) is incorporated in the GCF. The alive gross capital formation (GCFa) may incorporate own production factors indirectly through intermediate consumptions of grazing and harvested forage. The inanimate gross fixed capital formation (GFCFi) of infrastructures and livestock farming equipment is valued according to the respective production costs. Thus, we obtain an estimate of the net operating margin (NOM) of the livestock activity, which corresponds entirely to the manufactured capital of the case-study *dehesas*. This corollary is derived from the fact that the AAS avoids double-counting of the ecosystem services of own grazing (ESg) consumed by the livestock in the period, since these services are counted in the grazing activity, with this being the activity which produces them.

However, in the academic literature and government institutional reports, there are issues surrounding the polysemy of ecosystem services terms, since holistic definitions of the term are used which are incompatible with the concept of observed or simulated nature production factor transaction value applied in the AAS. The ecosystem services have been defined as "the direct and indirect contributions of ecosystems to human wellbeing, many of which do not have a market value and are ignored within evaluation [accounting] frameworks" ([3] p. 1361), being human wellbeing "a broad concept, one that includes many aspects of our everyday lives. It encompasses material wellbeing, relationships with family and friends, and emotional and physical health. It includes work and recreation, how one feels about one's community, and personal safety" (https://www.eopugetsound.org/science-review/section-3-nature-human-well-being, accessed on 11 February 2021). This academic definition of ecosystem services is equivalent to that of the United Nations Statistics Division (UNSD) which states that "ecosystem services (ES) are the contributions of ecosystems to benefits used in economic and other human activity" [13], being benefits defined as: "Goods and services that are ultimately used and enjoyed by people and which contribute to individual and societal wellbeing. Two broad types of benefits are described in ecosystem accounting—SNA benefits and non-SNA benefits" [13].

The livestock activity does not contribute ecosystem services to the observed or simulated exchange value of the total product (TP) of the livestock species in the period. We consider that the economic production function f of the livestock total product (TP) depends on the production factors of manufactured intermediate consumptions (ICm), labor (L), and manufactured fixed capital (FCm).

$$TP \equiv f(ICm, L, FCm). \tag{1}$$

Among other inputs of materials and services, the ICm contains the value of the grazing with observed or simulated market price. Apart from paid labor, L also includes self-employed family labor with simulated residual remuneration. The FCm comprises breeding (and working) livestock and inanimate fixed capital of infrastructures and equipment used in the livestock farming activity.

The accounting equation for the factorial distribution of TP incorporates the manufactured intermediate consumption (ICm), the labor cost (LC), the consumption of inanimate manufactured fixed capital (CFC), and the manufactured net operating margin

(NOM). The first three factors make up the total cost (TC) of the livestock species and the last production factor is the operating profit of the owner.

$$TP = ICm + LC + CFC + NOM. \tag{2}$$

$$TC = ICm + LC + CFC. \tag{3}$$

The official Farm Accounting Data Network (FADN) selects a list of standardized livestock commercial products (International Standard Industries Classification (ISIC)) and omits other noncommercial products, which are chosen and valued by the owner when making livestock investment decisions. The AAS methodology classifies the re-employed animal commercial raw materials as commercial intermediate products (IRMc), whereas some of these are recorded as final intra-consumption of the farm in the FADN methodology.

2.2.2. Livestock Noncommercial Intermediate Product of Private Amenity Self-Consumption

In our research applied to agro-silvo-pastoral extensive livestock systems, we verified the mixed commercial-amenity auto-consumption rationale of the large individual nonindustrial private landowners in real case studies [10,14,15].

The investments in livestock species which graze the Andalusian *dehesas* belonging to large nonindustrial private owners are motivated by the aim of obtaining normal net operating margins (NOMn). We assume the hypothesis that the owners will accept net operating margins at basic prices (NOM$_{bp}$) below the NOMn in exchange for ensuring the auto-consumption of private amenities, which would be lost if extensive rearing of their livestock species were abandoned. This voluntary opportunity cost reveals the implicit existence of a noncommercial intermediate production of private amenity service (ISSnca) of livestock farming.

In our articles subsequent to [16], we estimated the ISSnca for this type of livestock owner. Our hypothesis of the existence of ISSnca assumes that the nonindustrial owner obtains at least a normal net operating margin (NOMn), and a part of this margin may correspond to the ISSnca. Thus, in accordance with this hypothesis, by definition, all the livestock species that incur a net operating margin at basic price (NOM$_{bp}$) of less than the NOMn are seen to be compensated through the ISSnca, founded according to their voluntarily accepted opportunity costs (VOC).

The official FADN registers the NOM$_{bp}$ but does not allow the incorporation of the ISSnca when the NOM$_{bp}$ is less than the NOMn in the livestock investments. The AAS methodology, however, does incorporate the ISSnca and assumes that the ISSnca is used by the private amenity activity as the input of own ordinary noncommercial intermediate consumption of services (SSncooa) in the period [4,15,17].

In this study, we assume that the AAS estimates the ISSnca according to the difference between the NOMn and the NOM$_{bp}$.

$$ISSnca = NOMn - NOM_{bp}. \tag{4}$$

The NOM$_{bp}$ and NOMn are estimated directly. The NOM$_{bp}$ is estimated through the difference between the total product at basic price (TP$_{bp}$) and the total cost (TC). The NOMn is estimated by simulating a normal operating profitability rate (r) being obtained for the livestock farming immobilized manufactured capital (IMC).

$$NOM_{bp} = TP_{bp} - TC. \tag{5}$$

$$NOMn = r \times IMC. \tag{6}$$

In our application of the AAS to the case-study *dehesas* in Andalusia in this research, we estimated the NOMn applying a normal profitability rate (r) of 3%.

### 2.2.3. Intermediate Consumption

The AAS methodology extends the concept of own intermediate consumption to include own raw materials of animals and grazing (RMo) along with work in progress animals (WPu). Livestock rearing on the farm is considered that which is present for at least 6 months, while livestock grazing which remains for less than 6 months is considered as the final product of grazing. The FADN methodology omits own intermediate consumptions employed in obtaining livestock products in the period when they are not harvested and re-employed in the same period by the livestock farming activity (e.g., own grazed fodder). The livestock work in progress inventoried at the opening of the period and bought during the period is not included in the intermediate consumption by the FADN, which includes it in the product, forming part of the change in the animal inventory in the period.

The FADN methodology incorporates the purchases of inputs for use in the livestock activity as intermediate consumptions and records the period changes in the inventory of stored products in the total product. The AAS methodology only registers as purchases the external raw materials used regardless of the date when they were purchased and ignores the change in stored products in the period.

### 2.2.4. Forage Unit Livestock Consumption

The value of the grazing consumed (FUg) by livestock for each species present in the individual case-study farms is estimated through the residual valuation method. This involves calculating the difference between the total energy requirements (FUt) and the supplements (FUs) given to the animals in the period, measuring this quantity in forage units (FU), which refers to the energy content of a kilogram of barley with a humidity content of 14.1% and totals 2723 kcal [18].

$$FUg = FUt - FUs, \tag{7}$$

where g is grazing, t is total, and s is supplements.

The calculation of the FUt consumed by the livestock depends on the physical characteristics of the livestock population on the farm, as well as the gestation and lactation management parameters. Hence, the distribution and weights according to breeds and ages must be known. This information is gathered from the livestock inventories carried out at the beginning and end of the studied period. The method used for estimating these quantities is described in [19].

To estimate the amount of supplements during the year, data provided by the livestock owners with regard to the supplemented portion per species, together with the period during which this takes place, are used. All supplementary feed is transformed to FU content equivalent according to the type of feed [19].

Once the quantity of total forage units consumed by the livestock through grazing (FUg) in the period is known, it is possible to estimate the standing forage unit price. On the basis of the price paid for the annual lease of grazing pasture [4,19,20] and the total quantity of FUg consumed for each farm, the consumed standing forage unit price can be estimated.

The FU value of the pasture consumed by grazing livestock is considered a consumption of intermediate raw materials from the silvo-pastoral activity [4,15,17]. Grazing is only considered to have a value lower than or equal to the normal estimated for Andalusia for each vegetation type and province [19]. If the consumption of FUg exceeds the threshold, the excess FUg is considered to be free and, therefore, have a price of zero. Grazing consumption by game species is considered to be free as long as it is an open game reserve. In the case of closed game reserves, where the game species compete with domestic livestock for food, only the part of the consumption which allows the abovementioned normal consumption threshold to be reached is considered with economic exchange value, as long as this threshold has not been reached by the domestic livestock.

### 2.2.5. Total Income

We define the sustainable social income at social price ($TI_{sp,\,AAS}$) of the extensive livestock species production as the maximum possible consumption of the goods and services, with and without market prices, produced in the period and appropriated by the farmer without the final total capital of the livestock activity for the period decreasing in real terms ([14] p. 87). The indispensable detailed process of accounting records, which leads to the complete production and capital accounts of the AAS, gives the net value added at social price ($NVA_{sp,\,AAS}$) and the capital gain ($CG_{sp,\,AAS}$), respectively, which make up the total sustainable income of the livestock species.

$$TI_{sp,\,AAS} = NVA_{sp,\,AAS} + CG_{sp,\,AAS}. \tag{8}$$

### 2.2.6. Net Value Added

The net value added (NVA) represents the contributions of the labor cost (LC) and the net operating margin (NOM) services of the immobilized livestock capital to the value of the total product (TP) in the period. In other words, the NVA is the operating income embedded in the total product of livestock farming and does not incorporate the capital gains of the capital account. The net value added (NVA) of the livestock activity is estimated by the difference between the total product (TP), the intermediate consumption (IC), and the fixed capital at replacement cost (CFC).

$$NVA = TP - IC - CFC. \tag{9}$$

$$NVA = LC + NOM \tag{10}$$

The difficulty in estimating the individual economic result for the livestock species is the need to subjectively attribute the general shared costs and consumption of inanimate fixed capital. In the case of the costs, the criterion followed is that they should be divided among the same farm activities proportionally to the direct cost of each. The consumption of inanimate fixed capital is divided, in the case of infrastructures, according to the weight of the livestock units, and, in the case of equipment, the attribution is proportional to the time the equipment is used in the management of the individual species (Text S1, Supplementary Materials).

### 2.2.7. Capital Gain

The capital gains (CG) come from the breeding (and working) livestock and from the inanimate fixed capital of infrastructures (buildings) and equipment. The fixed capital account for the period records the revaluation (Cr), destructions due to death of the adult breeding livestock (Cda), and the final inanimate fixed capital (FCic), which is embedded in the effects of the depreciation (CFCi) and future revaluation/devaluation (CFCri), resulting from the effect of replacement cost change in the period ([16] Supplementary Text S11, p. 45). These records allow the livestock capital gain to be estimated as the Cr less the Cda plus the adjustment of the depreciation (Cadi). The latter comprises the consumption of fixed capital (CFCi) less its revaluation (CFCri) to avoid it being double-counted in the net value added ([4] Supplementary Text S1, pp. 4–5).

$$CG = Cr - Cda + Cadi. \tag{11}$$

$$Cadi = CFCi - CFCri. \tag{12}$$

The capital gains for animals of livestock species relate only to the breeding and working animals (CGa), and the capital gains of the inanimate capital (CGi) embrace the infrastructures and equipment employed in the management of the livestock species. The

CG shows the effects on the livestock farming incomes of the changes in the alive and inanimate fixed capitals of the extensive management of the livestock species.

$$CG = CGa + CGi. \tag{13}$$

In the AAS methodology, the sales of breeding and working (draught, denoting an animal used for pulling heavy loads) livestock fixed capital are incorporated in the capital gain only according to their revaluation in the period, as opposed to the FADN methodology which includes the sales of breeding and draught livestock in the total product. The revaluations of alive (CGra) and inanimate (CGri) manufactured fixed capitals employed in the productions of livestock species are incorporated in the capital gains (CG) estimated by the AAS. The destructions of livestock fixed capital (Cda) figure in the inventory implicitly changes in the total product of the FADN methodology. The AAS deducts the Cda from the revaluations (CGra) according to the opening inventory or purchase values in the period used for estimating the capital gain (CGa). Although the FADN methodology estimates the live fixed capital gain (CGa), it omits the inanimate fixed capital gain (CGi) of buildings and equipment. However, they are not completely omitted as the consumptions of inanimate fixed capital are incorporated in the net value added according to their replacement price ($CFCi_{rp}$).

The AAS methodology excludes Common Agricultural Policy (CAP) payment quotas from the fixed capital (FC) and it does not estimate the financial liabilities or the total capital (C) at the close of the period coinciding with the net worth (NW). The details for the distribution of inanimate fixed capital per species can be found in Text S1 (Supplementary Materials).

2.2.8. Net Operating Margins at Producer, Basic, and Social Prices

The FADN methodology estimates the net operating surplus/margin (NOM) at producer price (pp) and basic price (bp). In addition, the AAS methodology also estimates the NOM at social price (sp).

The normal net operating margin (NOMn) represents the lower limit of the net operating margin of the livestock farming, which is based on the hypothesis of voluntary opportunity cost of the livestock investment incurred in the period by the owners. The residual net operating margin at producer price ($NOM_{pp}$) is that which is derived from the observed and simulated market transactions of the livestock products generated in the period. The $NOM_{pp}$, which excludes the noncommercial intermediate products of services (ISSnc), is estimated by the difference between the total product at producer price ($TP_{pp}$) and the total cost at producer price ($TC_{pp}$) of the livestock species. The $NOM_{bp}$ is obtained when the FADN government compensations, reclassified in the AAS as the noncommercial intermediate product of compensation services (ISSncc), are added to the $NOM_{pp}$.

$$NOM_{pp} = TP_{pp} - TC_{pp}. \tag{14}$$

$$NOM_{bp} = NOM_{pp} + ISSncc. \tag{15}$$

In cases where the $NOM_{bp}$ exceeds the NOMn, the $NOM_{bp}$ and the $NOM_{sp}$ coincide.

$$NOM_{sp} = NOM_{bp}, \text{ if } NOM_{pb} \geq NOMn. \tag{16}$$

The FADN does not estimate the net operating margin at social price ($NOM_{sp}$) in cases where the net operating margin at basic price ($NOM_{bp}$) does not exceed the normal net operating margin (NOMn). In this situation, the AAS methodology incorporates the ISSnca leading to the valuation of the AAS net operating margin at social price ($NOM_{sp,AAS}$).

$$NOM_{sp,AAS} = NOM_{bp,AAS} + ISSnca, \text{ if } NOM_{pb,AAS} < NOMn. \tag{17}$$

2.2.9. Comparison of the FADN and AAS Net Values Added

Below, we describe the conceptual integration of the net values added under the Farm Accounting Data Network (FADN) and Agroforestry Accounting System (AAS) methodologies at basic prices and social prices. In the applications of the FADN, the estimates are not observable for livestock species as they are applied to the aggregate activities of the farm. The FADN methodology does not estimate the factorial distribution of the net value added of the livestock species among the production factors of labor and manufactured capital since the net mixed income ($NMI_{bp, FADN}$) is not separated into its two components of self-employed labor cost ($LCse$) and net operating surplus ($NOS_{bp,FADN}$). In contrast, the AAS methodology does present the factorial distribution of the net value added since it separates the net mixed income (NMI) into the components of imputed self-employed labor compensation ($LCse_{NMI}$) and normal manufactured net operating margin ($NOMn_{NMI}$) [4,10]. We refined the FADN in order to estimate the factorial distribution of the net value added of the farm under this system.

$$NVA_{bp,FADN} = LCse_{FADN} + NOS_{bp, FADN} + NMI_{bp, FADN}. \tag{18}$$

$$NVA_{bp,rFADN} = LC_{rFADN} + NOM_{bp, rFADN}. \tag{19}$$

The FADN methodology presents the aggregate results for the farm activities in which the total product of the livestock farming activity directed at a specific type of production predominates. Although the FADN does not present results for the livestock activity income of the farm, we make this income visible in order to compare the concepts of income from the extensive livestock activity with those of the Agroforestry Accounting System (AAS).

The official FADN methodology, unlike the academic AAS, does not incorporate the noncommercial intermediate product of government compensation services (ISSncc) and of private amenity self-consumption (ISSnca) in the total product (TP) of the livestock species. However, the FADN does incorporate the ISSncc in the net value added at basic price.

The variation in the value of the livestock species inventories net of livestock purchases in the period is considered in the FADN as a final product [5] (the FADN methodology also includes the change in stored inventory net of purchases which we omit in this comparison). The AAS does not register the net variation of the livestock inventories in the final product; rather, the value of the livestock census at the end of the current period (GWPCFa) and of the completed renewal of breeding livestock (GFCFa) are registered as final products of own-account gross capital formation (GCFa) in the period. At the opening of the period, the initial value of the livestock species work in progress inventories and the purchases of this type of livestock are registered in the AAS as a manufactured intermediate consumption cost of the period (WPmu). The previously mentioned livestock species inventory and purchase records in the production account of the AAS have the effect of excluding the capital gain of the breeding livestock (CGa) from the value added, in contrast to the FADN which does include it in the value added. However, the CGa is included by the AAS in the estimation of the total capital gain of the livestock activity (CG). The FADN does not count the own-account gross formation of inanimate fixed capital (GFCFi) of infrastructures and buildings and its corresponding total cost of inanimate investments (TCii) for the livestock species. The FADN excludes own grazing raw materials (RMog) from the livestock species intermediate consumption.

The differences in the results for the values added under the AAS and FADN methodologies are due to the net effects of the production account records for ISSnca, livestock inventories, own-account gross investments in infrastructures and buildings, and intermediate consumptions of own grazing raw materials.

$$NVA_{bp,FADN} = NVA_{sp,AAS} - ISSnca - GFCFi + RMog + TCii + CGa. \tag{20}$$

$$\text{GFCFi} = \text{TCii}. \tag{21}$$

$$\text{NVA}_{bp,FADN} = \text{NVA}_{sp,AAS} - \text{ISSnca} + \text{RMog} + \text{CGa}. \tag{22}$$

The livestock species net value added at basic price under the official FADN methodology ($\text{NVA}_{bp,FADN}$) can be considered an incomplete and inconsistent concept of the total income of the livestock activity at basic price. It is incomplete because it omits the intermediate consumption of grazing and the labor cost of own-account gross investment in infrastructures and buildings. This conclusion refers to the concept of net value added of the livestock activity, but not to its practice as it is not measured in the FADN. The aggregate value added of the national/sub-national products and of the farms does not incur the bias of incorporating grazing as it is only counted once in the livestock products and not as an intermediate product. The conceptual inconsistency of the FADN is due to the fact that it incorporates the fixed capital gain of breeding livestock.

The factorial distribution of the AAS net value added at social price ($\text{NVA}_{sp,AAS}$) among the labor cost ($\text{LC}_{AAS}$) and the net operating margin at social price ($\text{NOM}_{sp,AAS}$) is inevitably subjective in its components of self-employed work and the ISSnca. In the AAS valuation, self-employed work is rewarded with a maximum marginal hourly remuneration of 80% of the market remuneration for the same task done by employee labor [10]. Similarly, a subjective choice of normal profitability rate is necessary to estimate the ISSnca.

$$\text{NVA}_{sp,AAS} = \text{LC}_{AAS} + \text{NOM}_{sp,AAS}. \tag{23}$$

2.2.10. Profitability Rates

The operating profitability rates (Po) and capital gain (Pg) are estimated according to the ratios between the net operating margin (NOM) and the capital gain (CG) over the livestock immobilized capital (IMC) in the period. The total profitability rate (P) of the capital income (CI) of the livestock activity is estimated by the sum of both profitability rates.

$$\text{IMC} = \text{Co} + 0.5 \times (\text{Cb} + \text{TC} - \text{RMo} - \text{WPu} - \text{CFC} - \text{FPs} - \text{Cs}), \tag{24}$$

$$\text{Po} = \text{NOM/IMC}, \tag{25}$$

$$\text{Pg} = \text{CG/IMC}, \tag{26}$$

$$\text{P} = \text{CI/IMC}, \tag{27}$$

where Co is the opening capital, Cb is the capital bought (purchases), RMo is the own raw materials consumed in the production process, FPs are the final products sold, and Cs are the sales of capital occurring during the accounting year.

The Po rate is estimated subjectively, except where no self-employed labor is used and the net operating margin at basic price ($\text{NOM}_{bp}$) exceeds the simulated normal net operating margin (NOMn).

The livestock farm landowners risk their investment in livestock taking into account the overall result with other economic activities in the *dehesa*. However, the non-land-owning livestock farmers who lease the grazing land risk their investment taking into account only the results for the game species and ignoring the non-compensated effects of the livestock on the biophysical and economic results of the remaining activities on the farms which their livestock graze. In this research, the biases which may be incurred in the measurements of livestock profitability rates of large *dehesa* owners are canceled out in the exchanges among the activities linked to livestock, grazing, and private amenity.

Thus, the operating profitability rate at basic price ($Po_{bp,D}$) of the private activities of the *dehesa* as a whole is not affected by the incorporation of the noncommercial intermediate production of amenity services (ISSnca) of livestock species, coinciding with the operating profitability rates at basic price ($Po_{sp,D}$) and social price ($Po_{sp,D}$). The incorporation of the ISSnca of the *dehesa* owner has the effect of increasing and decreasing, by the same amount, the net operating margins of the livestock activity and private amenity, respectively [4,15,17]. Consequently, the operating profitability rates of the livestock farming at basic price ($Po_{bp,li}$) and social price ($Po_{sp,li}$) do not coincide. The latter results from adding the profitability rates of livestock species private amenity self-consumption ($Poa_{li}$) to the former.

$$Po_{bp,D} = Po_{sp,D}, \tag{28}$$

$$Po_{bp,li} \neq Po_{sp,li}, \tag{29}$$

$$Poa_{li} = ISSnca/IMCl_{li}, \tag{30}$$

$$Po_{spli} = Po_{bp,li} + Poa_{li}, \tag{31}$$

$$Po_{sp,li} = NOM_{sp,li}/IMCl_{li}, \tag{32}$$

where $IMC_{li}$ is the immobilized capital of the livestock species, and $NOM_{sp,li}$ is the net operating margin of the livestock species.

The estimates of the $Po_{bp,li}$ are objective in the case of the *dehesa* owners. The normal operating profitability rate ($Pon_{li}$) is not applicable in situations where the normal net operating margin of the livestock species ($NOMn_{li}$) is equal to or less than the net operating margin at basic price ($NOM_{bp,li}$); in this case, the ISSnca is zero and the $Po_{bp,li}$ and $Po_{sp,li}$ coincide, both being equal to or more than the $Pon_{li}$. As long as $NOMn_{li} \geq NOM_{bp,li}$, then the ISSnca is applicable and the $Po_{sp,li}$ and $Pon_{li}$ do not coincide.

### 2.2.11. Ordinary Cash Flow

The real management of the livestock species is conditioned by the monetary flows of revenues net of expenditures, which can lead to notably different values to those of the livestock species incomes in the same period. Although the ordinary cash flow (CFo) (including the annualized compensations and purchases of fixed capital, but not the incomes and payments of medium- and long-term loans received) does not signify direct income; instead, it shows the capacity of production at basic prices of the livestock species in the period in order to finance the costs of external production factors. The components of the monetary revenues (R) considered are the sales (S), commercial amenity self-consumption (Ac), and the noncommercial intermediate production of compensated services (ISSncc) of the government. The components of the expenditures (E) are the intermediate consumptions of the raw materials (RMb) and services (SSb) purchased, the head of livestock purchased (Cbli), employed labor cost (LCe), and inanimate consumption of fixed capital (CFC) in representation of ordinary bought inanimate fixed capital (FCb).

$$CFo = R - E. \tag{33}$$

$$R = S + Ac + ISSncc. \tag{34}$$

$$E = RMb + SSb + Cbli + LCe + CFC. \tag{35}$$

## 3. Results

In this section, we focus on describing the average absolute economic results with reference to livestock units in the case of the head of livestock along with the bee hives on the large nonindustrial privately owned individual case-study *dehesas* in Andalusia. However, the results for the livestock species are also aggregated in relative terms compared with the livestock farming activity and the private activities as a whole of the case-study *dehesa* owners.

The selected economic results for the livestock species estimated by applying the AAS methodology to the case-study *dehesas* are livestock species stocking rates, livestock species forage units grazed, livestock species units per annual work units, government compensation and private amenity noncommercial intermediate products of services, final product sales, bought and own commercial intermediate consumption of raw materials, consumption of inanimate fixed capital, net value added at basic and social prices, employee and self-employed labor costs, net operating margin at basic and social prices, total incomes at basic and social prices, operating cash flow at basic price, live and inanimate capital gains, and operating and total profitability rates at basic and social prices.

The notable effort required for daily monitoring of the time spent on tasks for each individual livestock species, in terms of both labor and machinery, may affect the lack of economic results published for overall productions in real case studies of silvo-pastoral and agroforestry farms such as the private *dehesas* in this study. The primary data for this research came from the RECAMAN (*Renta y Capital de los Montes de Andalucía*) project [15–17,20,21] (Additional information which readers may consider necessary to better understand the results of this research can be requested from the authors).

The absolute economic indicators for the livestock species are presented with reference to livestock units (LU) in the cases of ruminant species and horses; *Montanera* fattened head for Iberian pigs; sold head for suckling pigs; and hives in the case of bees. LU are estimated as a coefficient of the annual energy requirements of an empty "*Retinta*" cow with a weight of 450 kg [22]). A LU is equal to an annual requirement of 5171.32 Mcal of metabolizable energy (For meat cattle, sheep, and goats, it is equal to the equivalent LU of adult breeders. For fighting bulls and horses, it is the equivalent LU of animals older than 1 year). The values recorded in the production and capital accounts for the livestock activity are presented per hectare of the total aggregate of the case-study *dehesas*.

We organize the analysis of the production management and economic results by first presenting the eight classifications of species reared on the case-study *dehesas*, with two bovine and porcine variants. The second part of this section reveals the contribution of the livestock activity to the economy of the nonindustrial farmers of the large private *dehesa*s in the case study.

### 3.1. Livestock Species Production Management and Economic Results

#### 3.1.1. Meat Cattle

*Dehesas* with cattle, where production is directed toward calf rearing for sale after weaning at around 4–7 months, contributed 14.6% of the 0.44 LU/ha opening livestock stocking rate in the 21 case-study private *dehesas* in 2010 (Table 1). The native breeds reared are the "*Retinta*", the "Andalusian *Berrenda en Negro*" and the "Andalusian *Berrenda en Colorado*". The foreign breeds crossed with native cattle are the *Limousin* and the *Charolais*. There are also other foreign breeds such as the *Simmenthal-fleckvieh* (Table S1, Supplementary Materials). The labor employment ratio is 92.6 LU/AWU (annual work unit (AWU) is equivalent to 1826 h worked per year [23]) (Table 2 and Table A2, Appendix A). Meat cattle are those with the lowest dependence on supplementary feed, with grazing (FUg) accounting for 72.1% of the total consumption of metabolizable energy in 2010 (Table 3). However, there are cases where, despite free grazing of animals, the dependence on supplementary feed is very high due greater inclination of the owner toward recreational rather than livestock production activity (Table S2, Supplementary Materials). The

extensive management and coarseness of the grazing forage is reflected in the moderate ratios of births and sales of calves and in the high culling ratio, per breeding female (fb) (Table 4 and Table S2, Supplementary Materials).

**Table 1.** Livestock species stocking rate for large privately owned case-study *dehesas* in Andalusia (2010).

| Class | *Dehesas* (*n*) | Stocking Rate (LU/100 ha) |
|---|---|---|
| Meat cattle | 12 | 6.4 |
| Fighting bulls | 2 | 12.3 |
| Sheep | 8 | 3.0 |
| Goats | 6 | 1.6 |
| *Montanera* pigs | 9 | 10.0 |
| Extensive piglets | 1 | 0.0 |
| Horses | 8 | 2.1 |
| Total | 21 | 44.0 |

*Source*: Own elaboration. *Abbreviations*: *n* is the number of *dehesas* with the presence of this species more than 6 months per year (total = 21); LU is the livestock unit. *Notes*. A livestock unit (LU) is estimated as a coefficient of the annual energy requirements of an empty red cow with a weight of 450 kg [22]. An LU is equal to an annual requirement of 5171.32 Mcal of metabolizable energy. For meat cattle, sheep, and goats, it is equal to the equivalent LU of adult breeders. For fighting bulls and horses, it is equal to the equivalent LU of the animals older than 1 year. For *montanera* pigs, it is equal to the equivalent LU of the average number of Iberian pigs in *montanera* per year and *dehesa*. Absolute stocking LUs are as follows: meat cattle, 988 LU; fighting bulls, 1896 LU; sheep, 465 LU; goats, 241 LU; *montanera* pigs, 1536 LU; extensive piglets, 7 LU; horses, 330 LU. The total area of case-study *dehesas* is 15,372 hectares. The average area of case-study *dehesas* is 732 hectares.

**Table 2.** Livestock species annual labor and ownership for large privately owned case-study *dehesas* in Andalusia (2010).

| Class | Unit (u) | Livestock Owners (*n*) | Ratios | |
|---|---|---|---|---|
| | | | Labor (u/AWU) | Ownership (u/*n*) |
| Meat cattle (*n* = 12) | LU | 12 | 92.6 | 82.3 |
| Fighting bulls (*n* = 2) | LU | 2 | 137.0 | 947.9 |
| Sheep (*n* = 8) | LU | 8 | 111.4 | 58.1 |
| Goats (*n* = 6) | LU | 6 | 33.5 | 40.2 |
| *Montanera* pigs (*n* = 9) | heads [1] | 9 | 350.0 | 306.4 |
| Extensive piglets (*n* = 1) | heads sold | 1 | 162.9 | 93.0 |
| Horses (*n* = 8) | LU | 8 | 42.3 | 41.3 |
| Bees (*n* = 5) | hives | 5 | 586.5 | 138.0 |

*Source*: Own elaboration. *Abbreviations*: *n* is the number of *dehesas* with the presence of this species more than 6 months per year (total = 21); LU is the livestock unit; AWU is the annual work unit. *Notes:* [1] Average number of Iberian pigs in *montanera* per year and *dehesa*. A livestock unit is estimated as a coefficient of the annual energy requirements of an empty Retinta cow with a weight of 450 kg [22]. An LU is equal to an annual requirement of 5171.32 Mcal of metabolizable energy. For meat cattle, sheep, and goats, it is equal to the equivalent LU of adult breeders. For fighting bulls and horses, it is equal to the equivalent LU of the animals older than 1 year. Absolute unit measures are as follows: meat cattle, 988 LU; fighting bulls, 1896 LU; sheep, 465 LU; goats, 241 LU; *montanera* pigs, 2758 heads; extensive piglets, 93 heads; horses, 330 LU; bees, 690 hives. An annual work unit is equivalent to 1826 h worked per year [23]. The total area of case-study *dehesas* is 15,372 hectares. The average area of case-study *dehesas* is 732 hectares.

**Table 3.** Livestock species grazing forage unit consumption and supplements in large privately owned case-study *dehesas* in Andalusia (2010: %).

| Class | Grazing | | | | | | | | | Supplements | Total |
|---|---|---|---|---|---|---|---|---|---|---|---|
| | Grass and Browse | | | Acorn | | | Total | | | | |
| | Commercial | Free | Total | Commercial | Free | Total | Commercial | Free | Total | | |
| Meat cattle (*n* = 12) | 64.0 | 8.1 | 72.1 | | | | 64.0 | 8.1 | 72.1 | 27.9 | 100 |
| Fighting bulls (*n* = 2) | 43.5 | 3.8 | 47.3 | | | | 43.5 | 3.8 | 47.3 | 52.7 | 100 |
| Sheep (*n* = 8) | 46.4 | 5.4 | 51.8 | 3.2 | | 3.2 | 49.7 | 5.4 | 55.1 | 44.9 | 100 |
| Goats (*n* = 6) | 11.7 | 5.5 | 17.2 | 1.3 | | 1.3 | 13.0 | 5.5 | 18.6 | 81.4 | 100 |
| *Montanera* pigs (*n* = 9) | 14.9 | 0.5 | 15.4 | 15.4 | | 15.4 | 30.3 | 0.5 | 30.8 | 69.2 | 100 |
| Extensive piglets (*n* = 1) | | | | 15.7 | | 15.7 | 15.7 | | 15.7 | 84.3 | 100 |
| Horses (*n* = 8) | 19.4 | 0.3 | 19.8 | | | | 19.4 | 0.3 | 19.8 | 80.2 | 100 |
| Total (*n* = 21) | 34.0 | 3.7 | 37.7 | 5.1 | | 5.1 | 39.1 | 3.7 | 42.8 | 57.2 | 100 |

*Source*: Own elaboration. *Abbreviations*: *n* is the number of *dehesas* with the presence of this species more than 6 months per year (total = 21). The total area of case-study *dehesas* is 15,372 hectares. The average area of case-study *dehesas* is 732 hectares.

**Table 4.** Livestock yield ratios and prices for large privately owned case-study *dehesas* in Andalusia (2010).

| Class | Unit (u) | Baseline Unit (bl) | Ratio (u/bl) | Mean Price (EUR/u) |
|---|---|---|---|---|
| Birth | | | | |
| Meat cattle (*n* = 12) | he | fb | 0.6 | 277.6 |
| Sheep (*n* = 8) | he | fb | 1.1 | 52.9 |
| Goats (*n* = 6) | he | fb | 0.9 | 35.0 |
| Fighting bulls (*n* = 2) | he | fb | 0.6 | 226.9 |
| Sales | | | | |
| Meat cattle (*n* = 12) | he | fb | 0.5 | 474.3 |
| Sheep (*n* = 8) | he | fb | 1.0 | 53.2 |
| Goats (*n* = 6) | he | fb | 0.7 | 34.8 |
| *Montanera* pigs (*n* = 9) | arroba [1] | he [2] | 5.4 | 19.6 |
| Fighting bulls (*n* = 2) | he | fb | 0.4 | 1475.2 |
| Culling (breeders) | | | | |
| Meat cattle (*n* = 12) | he | fb | 0.2 | 491.8 |
| Sheep (*n* = 8) | he | fb | 0.1 | 21.9 |
| Goats (*n* = 6) | he | fb | 0.2 | 11.6 |
| Fighting bulls (*n* = 2) | he | fb | 0.1 | 143.1 |

*Source*: Own elaboration. *Abbreviations*: *n* is the number of *dehesas* with the presence of this species more than 6 months per year (total = 21); he is the number livestock heads; fb is the number of heads of female breeders. *Notes*: [1] Iberian pigs gain of weight during *montanera* fattening (1 arroba is 11.5 kg); [2] average number of Iberian pigs heads in *montanera* per year and *dehesa*. Absolute baseline unit measures are as follows: meat cattle female breeders, 813 heads; sheep female breeders, 2997 heads; goat female breeders, 1632 heads; fighting bull female breeders, 417 heads; *montanera* pigs, 2758 heads. The total area of case-study *dehesas* is 15,372 hectares. The average area of case-study *dehesas* is 732 hectares.

Of all the ruminant and equine species compared, the meat cattle activity is that which contributes the intermediate production of compensation services of greatest value per livestock unit (LU). In this respect, it is the third in terms of provision of self-consumed private amenity, sales of livestock products, and intermediate consumptions of raw materials and services purchased, per LU in all cases (Table 5 and Table A3, Appendix A). Own grazing per LU is also greatest and is 1.8 times greater than that of the next ruminant species (Table 5). It is the third species, whether ruminant or equine, in terms of use of inanimate capital investment per LU (infrastructure and equipment), as reflected by the inanimate fixed capital consumption value (amortization). This species also occupies third

place in the contribution to the net value added, labor, and net operating margin per LU (Table 5 and Table A3, Appendix A). It is second among the ruminant and equine species with regard to the intensity of immobilized capital (IMC) and capital losses (CG), in both cases per livestock unit. The CG losses are the consequence of adult livestock mortality and decreasing prices of livestock, buildings, and equipment in 2010 (Tables 5 and 6 and Table A4, Appendix A). The total income per LU is the third largest among the ruminant and equine species (Table 5). Furthermore, the ordinary cash flow is the third largest negative value (Table 5 and Table A5, Appendix A). The operating profitability rate at basic price is negative and, after incorporating the private amenities, reaches a value of 3.5% at social price (Table 7). The high volatility of livestock capital gain means that there is little point in drawing conclusions from the results of a single period (Table 7).

**Table 5.** Livestock species incomes and ordinary cash flows for large privately owned case-study *dehesas* in Andalusia under the Agroforestry Accounting System (AAS) (2010: EUR/u).

| Class | Meat Cattle (*n* = 12) (EUR /LU) | Fighting Bulls (*n* = 2) (EUR /LU) | Sheep (*n* = 8) (EUR /LU) | Goats (*n* = 6) (EUR /LU) | *Montanera* Pigs (*n* = 9) (EUR /head [1]) | Extensive Piglets (*n* = 1) (EUR/head sold) | Horses (*n* = 8) (EUR/LU) | Bees (*n* = 5) (EUR/hive) |
|---|---|---|---|---|---|---|---|---|
| Total product ($TP_{sp}$) | 1101.0 | 834.6 | 645.2 | 1241.4 | 877.7 | 366.3 | 2389.3 | 33.8 |
| Intermediate product ($IP_{sp}$) | 616.3 | 161.7 | 232.7 | 611.4 | 83.5 | 187.5 | 566.4 | 24.0 |
| Compensated (ISSncc) | 279.2 | 77.0 | 162.8 | 162.3 | 0.3 | | | 0.6 |
| Amenity auto-consumed (ISSnca) | 337.1 | 84.7 | 69.8 | 449.1 | 83.2 | 187.5 | 566.4 | 19.2 |
| Other intermediate product ($IPo_{PP}$) | | | | | | | | 4.2 |
| Final product ($FP_{PP}$) | 484.7 | 672.9 | 412.5 | 630.0 | 794.2 | 178.8 | 1822.9 | 9.9 |
| Sales ($FPs_{PP}$) | 216.5 | 146.8 | 339.2 | 489.8 | 446.1 | 125.7 | 20.0 | 8.8 |
| Gross fixed capital formation ($GFCF_{PP,cp}$) | 80.7 | 139.1 | 15.9 | 69.5 | 10.4 | 12.7 | 415.7 | |
| Gross work in progress formation ($GWPF_{PP}$) | 187.4 | 378.9 | 56.1 | 69.5 | 336.6 | 21.6 | 1387.2 | |
| Other final product ($FPo_{PP}$) | 0.0 | 8.1 | 1.3 | 1.1 | 1.1 | 18.8 | | 1.1 |
| Total cost ($TC_{PP}$) | 951.3 | 797.8 | 492.5 | 1162.9 | 844.1 | 354.2 | 2196.3 | 32.3 |
| Intermediate consumption ($IC_{PP}$) | 724.8 | 691.9 | 320.3 | 766.2 | 774.6 | 224.6 | 1766.6 | 23.1 |
| Bought ($ICb_{PP}$) | 403.7 | 192.1 | 230.6 | 596.0 | 246.5 | 69.6 | 592.5 | 18.9 |
| Own grazing and honey ($ICo_{PP}$) | 72.9 | 28.2 | 40.2 | 41.0 | 48.8 | 2.9 | 36.4 | 4.2 |
| Work in progress used ($WPu_{PP}$) | 248.2 | 471.6 | 49.4 | 129.1 | 479.3 | 152.1 | 1137.7 | |
| Inanimate consumption of fixed capital ($CFCi_{rp}$) | 90.3 | 13.0 | 56.4 | 128.3 | 13.3 | 18.3 | 125.8 | 8.5 |
| Net valued added at social price ($NVA_{sp}$) | 285.9 | 129.7 | 268.5 | 346.9 | 89.8 | 123.4 | 496.8 | 2.2 |
| Labor cost (LC) | 136.2 | 92.9 | 115.8 | 268.4 | 56.2 | 111.3 | 303.8 | 0.7 |
| Employee (LCe) | 136.2 | 92.9 | 88.7 | 268.4 | 56.2 | 111.3 | 302.7 | 0.7 |
| Self-employed (LCse) | 0.0 | | 27.1 | 0.0 | 0.0 | | 1.1 | 0.0 |
| Net operating margin at social price ($NOM_{sp}$) | 149.7 | 36.8 | 152.7 | 78.5 | 33.6 | 12.1 | 193.0 | 1.5 |
| Capital gain at producer price ($CG_{PP}$) | −169.9 | −44.4 | −100.3 | −303.2 | −22.0 | −4.5 | −41.3 | −10.0 |
| Alive (CGa) | −32.3 | −21.3 | −46.9 | −18.3 | −2.7 | 2.6 | 1.8 | |
| Inanimate (CGi) | −137.6 | −23.0 | −53.4 | −284.9 | −19.4 | −7.1 | −43.1 | −10.0 |
| Total income at social price ($TI_{sp}$) | 116.0 | 85.3 | 168.2 | 43.7 | 67.7 | 118.9 | 455.5 | −7.8 |
| Operating cash flow ($CF_{bp}$) | −146.5 | −50.3 | 144.0 | −327.0 | 50.5 | −43.5 | −562.3 | −17.7 |

*Source*: Own elaboration. *Abbreviations*: *n* is the number of *dehesas* with the presence of this species more than 6 months per year (total = 21); LU is the livestock unit. *Notes*: [1] Average number of Iberian pigs in *montanera* per year and *dehesa*. A livestock unit is estimated as a coefficient of the annual energy requirements of an empty Retinta cow with a weight of 450 kg [22]. An LU is equal to an annual requirement of 5171.32 Mcal of metabolizable energy. For meat cattle, sheep, and goats, it is equal to the equivalent LU of adult breeders. For fighting bulls and horses, it is equal to the equivalent LU of the animals older than 1 year. Absolute measures are as follows: meat cattle, 988 LU; fighting bulls, 1896 LU; sheep, 465 LU; goats, 241 LU; *montanera* pigs, 2758 heads; extensive piglets, 93 heads; horses, 330 LU; bees, 690 hives. The total area of case-study *dehesas* is 15,372 hectares. The average area of case-study *dehesas* is 732 hectares.

**Table 6.** Livestock species immobilized capital for large privately owned case-study *dehesas* in Andalusia (2010: EUR/u).

| Class | Unit (u) | Opening Capital 1 | Working Capital 2 | Immobilized Capital 3 = 1 + 2 |
|---|---|---|---|---|
| Meat cattle (*n* = 12) | EUR/LU | 4079.7 | 173.4 | 4253.1 |
| Fighting bulls (*n* = 2) | EUR/LU | 1163.9 | 63.7 | 1227.6 |
| Sheep (*n* = 8) | EUR/LU | 1990.0 | 32.1 | 2022.0 |
| Goats (*n* = 6) | EUR/LU | 2294.6 | 194.1 | 2488.7 |
| *Montanera* pigs (*n* = 9) | EUR/head [(1)] | 603.4 | 22.2 | 625.6 |
| Extensive piglets (*n* = 1) | EUR/head sold | 382.6 | 22.0 | 404.7 |
| Horses (*n* = 8) | EUR/LU | 5873.5 | 218.4 | 6091.9 |
| Bees (*n* = 5) | EUR/hive | 43.7 | 5.8 | 49.4 |

*Source*: Own elaboration. *Abbreviations*: *n* is the number of *dehesas* with the presence of this species more than 6 months per year (total = 21); LU is the livestock unit. *Notes*: [(1)] Average number of Iberian pigs in *montanera* per year and *dehesa*. A livestock unit is estimated as a coefficient of the annual energy requirements of an empty Retinta cow with a weight of 450 kg [22]. An LU is equal to an annual requirement of 5171.32 Mcal of metabolizable energy. For meat cattle, sheep, and goats, it is equal to the equivalent LU of adult breeders. For fighting bulls and horses, it is equal to the equivalent LU of the animals older than 1 year. Absolute measures are as follows: meat cattle, 988 LU; fighting bulls, 1896 LU; sheep, 465 LU; goats, 241 LU; *montanera* pigs, 2758 heads; extensive piglets, 93 heads; horses, 330 LU; bees, 690 hives. The total area of case-study *dehesas* is 15,372 hectares. The average area of case-study *dehesas* is 732 hectares.

**Table 7.** Livestock species profitability rates under the AAS for large privately owned case-study *dehesas* in Andalusia (2010: %).

| Class | Cattle Meat (*n* = 12) | Fighting Bulls (*n* = 2) | Sheep (*n* = 8) | Goats (*n* = 6) | *Montanera* Pigs (*n* = 9) | Extensive Piglets (*n* = 1) | Horses (*n* = 8) | Bees (*n* = 5) |
|---|---|---|---|---|---|---|---|---|
| $Po_{bp}$ | −4.4 | −3.9 | 4.1 | −14.9 | −7.9 | −43.3 | −6.1 | −35.8 |
| $Po_{sp}$ | 3.5 | 3.0 | 7.6 | 3.2 | 5.4 | 3.0 | 3.2 | 3.1 |
| $P_{bp}$ | −8.4 | −7.5 | −0.9 | −27.1 | −11.5 | −44.5 | −6.8 | −56.0 |
| $P_{sp}$ | −0.5 | −0.6 | 2.6 | −9.0 | 1.8 | 1.9 | 2.5 | −17.1 |

*Source*: Own elaboration. *Abbreviations*: *n* is the number of *dehesas* with the presence of this species more than 6 months per year (total = 21); Po is the operating profitability rate; P is the total profitability rate; subscript sp represents social price; subscript bp represents basic price. The total area of case-study *dehesas* is 15,372 hectares. The average area of case-study *dehesas* is 732 hectares.

### 3.1.2. Fighting Bulls

The production systems for native breeds of fighting bulls differ from those of the mother cows: the sale of calves for meat after weaning, rearing of renewal breeding females, and male offspring selected for fighting from the age of two up to a maximum of 5 years of age. It is precisely the greater presence of males selected for fighting and particularly their management (similar to that of other species with adult breeders in extensive regimes) which necessitates their inclusion in the estimate of livestock units (LU) in order to be able to compare the results with those for the management of other ruminant and equine species (Table 1). Thus, although only present on two of the case-study farms, they account for 28.0% of the opening livestock stocking units (Table 1), and the LU/AWU ratio is 1.5 times that of the meat cattle (Table 2 and Table A2, Appendix A). The grazed forage units made up 47.3% of the total consumption of metabolizable energy in 2010, which is the third highest grazing ratio of all the species reared on the case-study *dehesas* (Table 3). The average sale price of the animals is three times greater than that recorded for meat cattle due to the higher value of the males reared for bullfighting (Table 4). The culling rates are lower than for meat cattle, although it is not possible to draw firm conclusions in this regard from only two case studies (Table 4 and Table S3, Supplementary Materials).

The economic results for fighting bulls show similar trends to those for meat cattle (Tables 5–7 and Tables A3–A5, Appendix A). The important differences in sales of animals for bullfighting observed among the analyzed farms are due to the fact that, in one case,

the production of animals for bullfighting is regular and established, whereas, in the other case, renewals were incorporated to increase the size of the herd in the year 2010 (Table S3, Supplementary Materials). However, since the data only come from two farms, it is not possible to draw conclusions with regard to the indicators for the different productive aims of bovine livestock reared on the *dehesas.*

### 3.1.3. Sheep

*Dehesas* with ovine livestock, where production is directed toward lamb rearing for sale after weaning at around 1–2 months, account for a modest 6.9% of the opening livestock stocking units of the case study *dehesas* (Table 1). The native breeds are the "*Segureña*" and the "*Merina de Grazalema*", while the crosses of foreign and native are the "*Ile de France*" with "*Merina*" (Table S1, Supplementary Materials). The labor ratio is 111.4 LU/AWU (Table 2 and Table A2, Appendix A). Sheep for meat production is the species with the second lowest dependence on supplementary feed, with grazing making up 55.1% of the total forage unit consumption (Table 3). Extensive management of sheep presents birth ratios slightly above one, sales close to one, and a culling ratio of 0.1, in all cases per female breeder (fb) (Table 4 and Table S4, Supplementary Materials).

The ovine economic results are compared below with those for the rest of the ruminant and equine species. The ovine for meat production is the second highest contributor of intermediate production of compensation services per livestock unit (LU) of all the ruminant and equine species in the case studies. With regard to provision of self-consumed private amenity services, the contribution of this species is the lowest, whereas it is the second highest for sales of livestock products and the fourth in terms of intermediate consumptions of raw materials and services purchased (Table 5 and Table A3, Appendix A). The consumption of own pasture per livestock unit is the third highest, although very close to that of goats and horses (Table 5 and Table A3, Appendix A). Similarly, of the ruminant and equine species, it is the fourth in terms of use of inanimate capital investment per LU (infrastructure and equipment), as reflected by the inanimate fixed capital consumption (amortization). It also occupies fourth place in contributions of net value added, as well as labor, and second place in terms of net operating margin per livestock unit (Table 5 and Table A3, Appendix A). Sheep occupy fourth place in intensity of immobilized capital (IMC) and third place in losses of capital (CG) per livestock unit, for the same reasons as those mentioned above for bovine livestock (Tables 5–6 and Table A4, Appendix A). The total income per LU is second highest among the ruminant and equine species (Table 5). The ordinary cash flow is notably positive, which is explained by the fact that this species for which self-employed labor is most used in the case-study *dehesas* (Table 5 and Table A5, Appendix A). The operating profitability rates at basic prices and social prices are notably positive, reaching values of 4.1% and 7.6%, respectively (Table 7).

### 3.1.4. Goats

*Dehesas* with caprine livestock, where production is directed toward both milk and goat kid breeding for sale after weaning at around 5 weeks, make up 3.6% of the opening livestock stocking units of the case-study *dehesas* (Table 1) with a labor ratio of LU/AWU of 33.5 (Table 2 and Table A2, Appendix A). Goats for both suckling and milk production have the second highest dependency on supplementary feed, with grazing only making up 18.6% of their total consumption of forage units (Table 3). Extensive management of goats presents ratios for births of around one and sales of kids of 0.7, while the culling ratio is 0.2. per female breeder (Table 4 and Table S5, Supplementary Materials). The native breeds are the "*Granaina*", the "*Malagueña*", the "*Blanca Andaluza*", the "*Serrana*", and the "*Murciano-Granadina*". There are also crossbreeds with no specific genealogical ascription (Table S1, Supplementary Materials).

Among the ruminant and equine species in the case studies, the mixed meat/milk production goat has the third highest intermediate production of compensation services per LU (very close to that of the ovine species) (Table 5). If other economic indicators per

LU are compared among these species, goats present the second highest value for provision of self-consumed private amenity services, as well as the highest in terms of sales of livestock products and the purchase of intermediate consumption of raw materials and services (Table 5 and Table A3, Appendix A). The consumption of own grazing per LU for this species is the fourth lowest (Table 5 and Table A3, Appendix A). Of the ruminant or equine species, the goat is that with the greatest use of inanimate capital investments per LU (infrastructure and equipment). It is also the second in net value added and labor. Due to the weight of the employed labor, it is the species (ruminant or equine) with the second lowest net operating margin per livestock unit (Table 5 and Table A3, Appendix A). It occupies third place with regard to intensity of immobilized capital (IMC) and first place in capital losses (CG), which is explained by the drop in goat prices in Andalusia in the year 2010, per LU in both cases (Tables 5 and 6 and Table A4, Appendix A). The total income per LU for this species is the lowest of all the ruminant and equine species as a consequence of the losses of capital (Table 5). The ordinary cash flow is the second largest negative value per LU in the case-study *dehesas* (Table 5 and Table A5, Appendix A). The operating profitability rate at basic prices is markedly negative, reaching a positive value of 3.2% at social prices (Table 7).

3.1.5. *Montanera* Iberian Pig Fattening

*Dehesas* with Iberian pigs, where production is directed toward *montanera* fattening (during the period from October to January, the *montanera* is the feeding and fattening of the Iberian porcine livestock without using supplementary feed) over a 4 month period with acorns mainly from holm oaks, make up 22.7% of the opening livestock stocking units of the case-study *dehesas* (Table 1 and Tables S6 and S7, Supplementary Materials) with a labor ratio of 350 head fattened per AWU (Table 2 and Table A2, Appendix A). Among the species studied, the Iberian pig for *Montanera* fattening is the fourth most dependent on supplementary feed, with grazing accounting for 30.8% of the total consumption of forage units. As would be expected, it is also the species with the greatest acorn consumption (Table 3 and Table A1, Appendix A). In the case-study *dehesas*, the extensive management of *Montanera* fattened pigs is carried out with a small number of female breeders, or Iberian suckling pigs are purchased for fattening, with both management models being compatible. The heads of fattened Iberian pigs gain 5.4 @/he (an arroba (@) is equal to 11.5 kg.) over the *montanera* period (Table 4 and Tables S6 and S7, Supplementary Materials).

The average weighted price per forage unit (grass, browse, and acorns) on the private *dehesa* farms grazed by the ruminant livestock species is 0.07 EUR/FU, varying from 0.02 EUR/FU to 0.23 EUR/FU. The average weighted price per forage unit grazed in the *montanera* (comprising acorns and a part no greater than one-third made up of grasses and micro wildlife) by Iberian pigs (pure breeds *Negro Entrepelado* and *Lampiño*, and/or crosses with the Duroc breed) on the private *dehesas* is 0.18 EUR/FU. The average weighted price for *montanera* fattening of the Iberian pigs is 19.6 EUR/@ with a consumption of 2.5 FU/day per animal (Table 4 and Table S7, Supplementary Materials). On the case-study private *dehesas*, the fee for Iberian pig fattening attributed to the *montanera* accounts for 34% of the value of the *montanera* fattened pigs.

To avoid distortions caused by specific aspects of *montanera* pig management, as well as the nutritional and metabolic characteristics, we consider it more appropriate to estimate the economic values on the basis of the heads of pigs that remain in the *montanera* regime for the whole year. This allows us to make an equivalent comparison of the management, per livestock unit, of the ruminant and equine livestock on the case-study *dehesas*. *Montanera* pigs contribute the lowest intermediate production of compensation services, taking into account the fattened animals, corresponding exclusively to culling of unhealthy animals (Table 5 and Tables A3 and A6, Appendix A). According to this comparison, this species presents the second lowest value for provision of self-consumed private amenity services, the second highest value for livestock product sales, and the fourth

highest for purchases of intermediate raw material and service consumption (Table 5 and Table A3, Appendix A). In comparison with the other livestock species, the consumption of own grazing is the second highest (Table 5). With regard to the use of inanimate capital investments (infrastructures and equipment), *montanera* pigs are the second lowest of the livestock species. Furthermore, this species presents the lowest net values added and labor (Table 5 and Table A3, Appendix A). It is also the species which presents the lowest net operating margin, immobilized capital (IMC), and losses of capital (CG) (Tables 5 and 6 and Tables A3 and A4, Appendix A). The total income per head of fattened pigs is the second lowest (Table 5). The ordinary cash flow is the second highest positive value among the species compared in the case-study *dehesas* (Table 5 and Table A5, Appendix A). The operating profitability rate at basic prices is notably negative, while, at social prices, it reaches a positive value of 5.4% (Table 7).

### 3.1.6. Extensive Piglet Production

Although we present the results for extensive production (with ecological certification) of suckling pigs of the *Negro Entrepelado* Iberian pig breed for illustrative purposes, we do not describe this production as it only has a token presence on one of the case-study *dehesas* (Tables 1–7, Tables A3–A5, Appendix A, and Tables S1 and S8, Supplementary Materials).

### 3.1.7. Horses

*Dehesas* with equine species, directed toward the breeding of these species for use as studs and recreation or work horses, make up 4.9% of the opening livestock stocking units, with a ratio of 42.3 LU/AWU (Tables 1 and 2). It is important to note that, as with the fighting bulls, animals more than 1 year old are incorporated in the measurement of the LU, which is due to the fact that their extensive management would be equivalent to that of adult breeders of other ruminant species (Table 1). The native breeds are the *Hispano-Arabe* and Andalusian pure breed. There are also foreign breed Anglo-Arabic horses, as well as mixed-breed mules, ponies, horses, and donkeys with no defined genealogical ascription (Table S1, Supplementary Materials). The equine species are the second most dependent on supplementary feed, with grazing making up 19.8% of their total consumption of metabolizable energy in 2010 (Tables 3 and 4 and Table S9, Supplementary Materials).

Among the ruminant and equine species, the latter are those which account for the greatest intermediate production of self-consumed private amenity services per LU (Table 5 and Table A3, Appendix A). This is coherent with the predominantly recreational use of the case-study *dehesas*. As a consequence, they are also the species with the lowest sales. They are first per LU, in terms of both the gross capital formation (given that the objective of the farms is the production of animals for recreational purposes, workhorses, or studs) and the intermediate consumption of raw materials and services purchased (Table 5 and Table A3, Appendix A). The consumption of own grazing per LU is the second lowest of the species considered (Table 5). Equine species are the second largest consumers of inanimate fixed capital per LU, with the values being similar to those for goats (amortization). They are first, by a considerable distance in all cases, with regard to contributions of net value added, labor, net operating margin, immobilized capital (IMC), and total income per LU (Tables 5 and 6 and Table A4, Appendix A). They also present the lowest capital losses (CG) of the species compared per LU due to the absence of adult livestock deaths and greater stability of prices for this type of animal. However, the value of the CG is still negative due to falls in the prices of buildings and equipment in 2010 (Table 5 and Tables A4 and A7, Appendix A). The ordinary cash flow presents the greatest negative value per LU due to the accumulation of productions for the year (Table 5 and Table A5, Appendix A). The operating profitability rate at basic price is negative and, after incorporating the private amenities, reaches a value of 3.2% at social price (Table 7). The high volatility of the capital gain of the livestock farming activity means that it is pointless to attempt to identify a trend for the total profitability rate on the basis of results for just one period.

### 3.1.8. Bees

*Dehesas* with apiculture activity present a labor ratio of 586.5 hives/AWU (Table 2 and Table S10, Supplementary Materials). The economic values per hive and period (year) are not comparable with the rest of the activities which are expressed per LU or head of adult animals (Table 5). Own intermediate consumption of honey is the only value recorded which is different from the grazing of the rest of the species (Table 5). The net value added and the net operating margin are slightly positive, while the total income and the ordinary cash flow are both negative (Table 5 and Table A3, Appendix A). The operating profitability rate at basic price is notably negative, while, at social price, it has a positive value of 3.1% (Tables 6 and 7 and Table A4, Appendix A).

### 3.2. Large Private Dehesa Livestock Activity Economies

### 3.2.1. Comparison of Economic Indicators for Livestock Species and Activity

The total production for the livestock species increases in relation to the commercial production due to the amount of noncommercial intermediate products of compensation and private amenity services incorporated (Table 5 and Table A3, Appendix A).

Government compensations (ISSncc) are based on the landscape activity conservation service additional final products which they generate. Thus, the ISSncc of the livestock species are considered inputs of own ordinary noncommercial intermediate consumption of compensated services (SSncooc) of the public landscape activity. The incorporation of the SSncooc increases the final productions consumed of the public landscape activity by the amounts of the former; therefore, the net operating margin (NOM) and the net value added (NVA) of the landscape activity are not affected. The bovine and ovine livestock species are those which present the greatest ISSncc values (Table 8 and Table A6, Appendix A).

**Table 8.** Comparison of selected economic indicators for livestock species on large privately owned case-study *dehesas* in Andalusia (2010: %).

| Class | Meat Cattle/ Livestock Activity (*n* = 12) | Fighting Bulls/ Livestock Activity (*n* = 2) | Sheep/ Livestock Activity (*n* = 8) | Goats/ Livestock Activity (*n* = 6) | *Montanera* Pigs/ Livestock Activity (*n* = 9) | Extensive Piglets/ Livestock Activity (*n* = 1) | Horses/ Livestock Activity (*n* = 8) | Bees/ Livestock Activity (*n* = 5) | Livestock Activity/*Dehesa* Private Activities (*n* = 21) |
|---|---|---|---|---|---|---|---|---|---|
| Government compensation | 51 | 27 | 14 | 7 | 0 | | | 0 | 92 |
| Private amenity | 31 | 15 | 3 | 10 | 21 | 2 | 17 | 1 | 39 |
| Final product sales | 11 | 14 | 8 | 6 | 61 | 1 | 0 | 0 | 55 |
| Bought intermediate consumption | 21 | 19 | 6 | 8 | 36 | 0 | 10 | 1 | 66 |
| Own intermediate consumption | 24 | 18 | 6 | 3 | 44 | 0 | 4 | 1 | 24 |
| Consumption of inanimate fixed capital | 35 | 10 | 10 | 12 | 14 | 1 | 16 | 2 | 36 |
| Net value added at social price | 24 | 21 | 11 | 7 | 21 | 1 | 14 | 0 | 19 |
| Employee labor | 20 | 26 | 6 | 9 | 23 | 2 | 15 | 0 | 42 |
| Self-employed labor | 0 | | 97 | 0 | 0 | | 3 | 0 | 46 |
| Net operating margin at social price | 32 | 15 | 15 | 4 | 20 | 0 | 14 | 0 | 11 |

*Source*: Own elaboration. *Abbreviations*: *n* is the number of *dehesas* with the presence of this species more than 6 months per year. The total area of case-study *dehesas* is 15,372 hectares. The average area of case-study *dehesas* is 732 hectares.

The self-consumption of private amenities (ISSnca) exceeds, on average, the government compensations (ISSncc) for livestock species on the case-study private *dehesas* (Table 8 and Table A6, Appendix A). The livestock species with the greatest ISSnca value per hectare, on average, is the meat bovine, followed by the porcine livestock (Table 8 and Table A6, Appendix A). The lowest ISSnca value estimated is that for the bees (Table 8

and Table A6, Appendix A). The own ordinary noncommercial intermediate consumptions of private amenity (SSncooa) lead to a reduction in the net operating margin and in the net value added of the private amenity activity by the amount of the former.

The livestock activity accounts for 92% and 39% of the ISSncc and ISSnca, respectively, of the private activities of the case-study *dehesas* (Table 8 and Table A6, Appendix A). The sale of final livestock products excludes the animals whose main function is breeding, except for the porcine breeders which are reclassified during the period as work in progress for fattening. *Montanera* fattening of Iberian pigs accounts for 61% of the livestock sales, and the livestock activity makes up 55% of the sales for the overall private activities of the case-study *dehesas* (Table 8 and Table A6, Appendix A).

The bovine and *montanera* fattened Iberian porcine species account for 76% of the intermediate consumption purchased of the livestock activity. This activity makes up 66% of the total intermediate consumptions purchased for the overall activities of the case-study *dehesas* (Table 8 and Table A6, Appendix A).

The consumption of grazing makes up 99% of the livestock activity own intermediate consumption and much of that, 44%, corresponds to the *montanera* fattened Iberian pigs. Own intermediate consumptions of grazing and honey account for 24% of overall own intermediate consumption of the activities on the case-study *dehesas* (Table 8).

The bovine and *montanera* fattened Iberian porcine species account for 59% of the consumption of inanimate fixed capital of the livestock activity. The latter makes up 36% of the total inanimate fixed capital consumption of the overall activities on the case-study *dehesas* (Table 8 and Tables A6 and A7, Appendix A).

The bovine and *montanera* fattened Iberian porcine species account for 67% of the net value added at social price ($NVA_{sp}$) of the livestock activity. Thus, the livestock activity makes up 19% of the total net value added of the farmer activities as a whole on the case-study *dehesas* (Table 8 and Table A6, Appendix A). This relatively low contribution of the livestock activity is due to the fact that large nonindustrial private owners prioritize self-consumption of the final products of private amenity registered in the private amenity activity and the net value added of the cork activity.

The private nonindustrial livestock farmers of the large private *dehesas* generally manage their livestock species using employed labor. The latter is mainly concentrated in the bovine and *montanera* fattened Iberian porcine species, making up 68% of the total for the livestock activity, while the latter accounts for 42% of the total for the livestock activity as a whole on the case-study *dehesas* (Table 8 and Table A6, Appendix A). In the studied *dehesas* the contribution of self-employed labor is negligible (Tables A2 and A6, Appendix A), with 97% of this type of labor corresponding to the ovine livestock activity (Table 7), which in turn accounts for 46% of the total self-employed labor for the activities as a whole on the case-study *dehesas* (Table 8 and Table A6, Appendix A).

The bovine and *montanera* fattened Iberian porcine species account for 52% of the net operating margin at social price ($NOM_{sp}$) of the livestock activity. Furthermore, 11% of the net operating margin value for the farmer activities as a whole on the case-study *dehesas* corresponds to the livestock activity (Table 8 and Table A6, Appendix A).

### 3.2.2. Comparison of Net Values Added for Livestock Activity under the FADN and AAS Frameworks

The AAS methodology applied to these case-study *dehesas* in Andalusia incorporates the noncommercial intermediate product of the private amenity service (ISSnca) self-consumed by the individual nonindustrial owners along with the intermediate consumption of own grazing (RMog) in the livestock activity. The official FADN methodology omits the estimation of the ISSnca and RMog, and it incorporates the livestock capital gain (CGa) in the final product of the inventory change. Although the FADN does not present results for each livestock activity, but rather estimates the net value added at basic price ($NVA_{bp,FADN}$) for the farm activities as a whole, classified according to the main technical-economic orientation, we simulated its estimation of the livestock activity on the basis of



production and capital balance account records of the AAS (Tables A6 and A7, Appendix A). Applying Equation (22) provides an estimate of the FADN value added of the livestock activity at basic price (NVA$_{bp,FADN,li}$) of 23.8% of that estimated by the AAS at social price (NVA$_{sp,AAS}$) (Table 9).

**Table 9.** Livestock species and activity net value added under the refined Farm Accounting Data Network (FADN) of large privately owned *dehesas* in Andalusia (2010: EUR/ha).

| Class | Meat Cattle (*n* = 12) | Fighting Bulls (*n* = 2) | Sheep (*n* = 8) | Goats (*n* = 6) | *Montanera* Pigs (*n* = 9) | Extensive Piglets (*n* = 1) | Horses (*n* = 8) | Bees (*n* = 5) | Livestock (*n* = 21) |
|---|---|---|---|---|---|---|---|---|---|
| AAS Net valued added at social price (NVAsp,AAS) | 18.4 | 16.0 | 8.1 | 5.4 | 16.1 | 0.7 | 10.7 | 0.1 | 75.6 |
| Amenity auto-consumed (ISSnca) | 21.7 | 10.4 | 2.1 | 7.1 | 14.9 | 1.1 | 12.2 | 0.9 | 70.4 |
| Own grazing raw material (RMog) | 4.7 | 3.5 | 1.2 | 0.6 | 8.8 | 0.0 | 0.8 | | 19.6 |
| Alive capital gain (CGa) | −2.1 | −2.6 | −1.4 | −0.3 | −0.5 | 0.0 | 0.0 | | −6.8 |
| FADN Net valued added at basic price (NVAbp,FADN) | −0.7 | 6.4 | 5.8 | −1.2 | 9.5 | −0.4 | −0.7 | −0.8 | 18.0 |

*Source*: Own elaboration. *Abbreviations*: *n* is the number of *dehesas* with the presence of this species more than 6 months per year. The total area of case-study *dehesas* is 15,372 hectares. The average area of case-study *dehesas* is 732 hectares.

## 4. Discussion

In this section, we discuss the economic implications, the strengths and weaknesses of the AAS application, and policy implications which can be derived from the livestock accounting framework applied to the case-study *dehesas* in Andalusia.

### 4.1. Economic Implications

The investment in ruminant livestock to produce animals for meat or breeding up to the weaning of offspring and/or for milk production is not, at market prices, a competitive investment in terms of the profit at basic price obtained by the livestock owner. Our hypothesis of obtaining a minimum normal implicit operating profitability rate of 3% eliminates the uncertainty of the operating profitability of the livestock species. The operating profitability rates at basic prices (Po$_{bp}$) for the livestock species estimated are usually below the normal rates assumed of 3% and commonly may even be negative (Table 7).

Having assumed that the values of the private amenity services (ISSnca) for the livestock species are the same as the differences between the normal net operating margins (NOMn) and the basic net operating margins (NOMb) means that the operating profits at social price (Po$_{sp}$) will always be equal to or greater than 3% (Table 7). This finding for livestock farmers who continue investing in the rearing of livestock species allows us to understand why some might abandon these species, along with the decline in new livestock farmers entering the activity. This situation may be due, on the one hand, to greater requirements for profitability at basic prices and, on the other, to self-consumption of private amenities below the commercial opportunity costs accepted voluntarily by the owners of the livestock species.

The findings reveal that, under normal conditions, the nonindustrial private *dehesa* owners rear meat cattle at noncompetitive basic prices and that it is the additional noncommercial benefit which they receive in the form of self-consumed private amenities associated with the rearing of their livestock which allows them to achieve a competitive operating profit. The ordinary cash flow, which is notably negative, confirms the noncommercial benefit which is omitted in the measurement of the profitability rate at basic price. However, it is recognized in the profitability rate at social price and, therefore, is a

more reliable reflection of the real profitability obtained by the farmer from the rearing of meat cattle on their private *dehesa*.

The capital gain profitability rates for the livestock species given steady state of prices and livestock inventory tend toward zero; therefore, the price variations of fixed capital in the period together with the deaths of breeding and working livestock are likely to be reflected in relatively high volatility of values between consecutive periods. In 2010, there was a notable decrease in the prices of inanimate fixed capital investments (infrastructures and equipment). The revaluation of the fixed capital of breeding and working livestock was slightly above the value of the losses (deaths) of these animals in 2010 (Table 5 and Tables A4 and A7, Appendix A). However, the high volatility of the inter-period variations in livestock capital gain—originating in the historical investments in inanimate fixed capital of infrastructures and equipment—can lead to total profitability rates for the livestock species below the rate which we assume as normal in alternative non-livestock commercial asset investments.

We estimated the total profitability rates at basic prices ($P_{bp}$) and social prices ($P_{sp}$) for the livestock species, although we do not believe that they have a significant influence on the decisions of livestock owners with regard to their investments (Table 7). Other explanations for the decline in full-time or majority dedication of self-employed family labor to the livestock activity in the large case-study *dehesa* farms are of relevance. Among these explanations is the expansion of large game hunting on the private farms for both commercial and self-consumption motivations of the private landowners.

*4.2. Strengths and Weaknesses of the AAS*

The official FADN methodology omits the estimation of values added and margins of the individual livestock species and of the livestock farming activity itself since the results for values added at basic prices of the economic activities are aggregated according to the technical-economic orientation of the farms (Table 10). Hence, our analysis of the income estimates at social prices for the individual livestock species can be considered a conceptual novelty (Table 10). Moreover, the results are consistent with the theory of sustainable total income in this application of the AAS methodology to the case-study private *dehesas* (Table 10).

**Table 10.** Strengths and weaknesses of the AAS applied on large privately owned case-study *dehesas* in Andalusia.

| Strengths | Weaknesses |
|---|---|
| • It estimates the values added and margins of each individual livestock species and of the livestock farming activity itself. | • Sustainability depends on expected future events. |
| • It estimates income at social prices for the individual livestock species. | • The uncertainty of the hypothesis applied to estimate amenity service through a subjective competitive profitability rate. |
| • It is consistent with the theory of sustainable total income. | • The government expenditures on healthcare of all the livestock species are not considered. |
| • The uncertainty of the hypothesis applied to estimate amenity service does not affect the net operating margin at social price of the individual dehesa activities as a whole. | • The depreciation of equipment and buildings for each of the livestock species is done by subjective criteria. |

*Source*: Own elaboration.

The biophysical sustainability of a natural area requires the safe minimum standard (SMS) thresholds of the unique natural variety to be considered at the close of the period [24], and the economic sustainability depends on the real total income of the natural space not declining. The weakness of this economic sustainability concept is that it depends on future events, since efficient economic grazing on landscapes of the *Quercus* genus, such as holm oak and cork oak, by definition, relies on the regeneration of these landscapes

which can only occur through programmed temporal absence of the grazing activity. If the opportunity cost incurred due to grazing being subject to programmed regeneration of the woodland exceeds the ISSnca, then the difference up to the cost incurred could be considered an intermediate production of services compensated by the government (ISSncc).

In the case-study farms, the woodland inventories were modeled on information from the third National Forest Inventory (NFI3) [25], and the natural renovation/plantation is scheduled for the end of the prescribed cycle of each generation of trees. Consequently, by definition, given these modeled future schedules, the livestock activities on the case-study farms are expected to be biophysically and economically sustainable. In other words, despite the dependence on future effects, this sustainability depends on expected future events (Table 10).

The uncertainty of the hypothesis applied in the AAS methodology, where each of the individual livestock species j reaches at least a normal net operating margin (NOMnj), does not affect the net operating margin at social price (NOM$_{sp,D}$) of the individual *dehesa* activities as a whole, although the incorporation of the ISSnca does affect the margins of the livestock and private amenity activities oppositely in accordance with the volume of the former (Table 10).

One shortfall of this research is that we did not consider the healthcare of all the livestock species compensated by the government (Table 10). Government compensations were, however, incorporated for the slaughter of certain species to control the spread of disease, such as that associated with porcine livestock. Not all costs related to vaccinations and other government campaigns to prevent or control disease were incorporated, although such measures may involve substantial amounts. In cases where these compensations were incorporated, we took into account the prevention and fight against certain endemic livestock diseases, undertaken directly by the government as a public service of the landscape conservation activity. Thus understood, recording livestock healthcare with direct public spending cost could be simulated as noncommercial intermediate production of service compensated (ISSncc) of the livestock activity. It would be necessary to simultaneously record this amount in the public landscape activity as an input of own ordinary noncommercial intermediate service consumption (SSncooc). The livestock capital balance account would register the withdrawals according to whether they are sales, uses, destruction, or other livestock withdrawals. The total social income would be reduced by the aggregate amount of the public spending and the change in the livestock total income. In conclusion, health damage positively affects the intermediate production and negatively affects the capital gain of the livestock activity.

Another existing limitation is the distribution of equipment and buildings for each of the livestock species, since it is common that, where there is more than one livestock species, their use may be shared, their use may vary, or the time utilized by the different species may not be clearly delimited (Table 10). To attenuate this limitation, in those cases where the use of inanimate capital is not delimited, a coefficient was estimated using the primary data collected in [21], according to the total food consumption, in forage units (FU), for each species and farm. The methodological details of this operation are provided in Text S1 (Supplementary Materials).

### 4.3. Policy Implications

The income results for the livestock species estimated by applying the Agroforestry Accounting System (AAS) cannot be derived from the application of the official FADN. This is due to the fact that this methodology does not admit the incorporation of the ISSnca, as well as to the substitution of the valuation of the final product of the private amenity service for the production cost (without margin) in the FADN, instead of the simulated exchange value which is applied in the AAS, derived from the willingness to pay declared by the owners.

The farms in the case study are located in rural areas with low population densities. In this context, livestock farming contributes in relative terms to a relevant part of the employment generated on the farms and to a population establishment with strong local roots. This population attracts recreational visitors with offers of rural accommodation, local food, and craft products. Extensive livestock farming, thus, contributes to the maintenance of the traditional cultural landscape and to the biophysical and economic sustainability of the territory in rural areas.

The biggest threat to extensive livestock farming is the lack of shepherds for sheep and goat species. The increased demands of herders on site by the small livestock species, during grazing time, hamper improvements in labor productivity. Access to public health and education services for children also suffers. In this context, the replacement of sheep and goats by cows and big game species is an ongoing process that threatens the continuity of this type of livestock farming, which is key to the use of pastures in the arid and semiarid lands of the case study areas and, in general, of extensive livestock farming in the Mediterranean climate regions of the Iberian Peninsula.

The right to grazing consumption under conditions which degrade the natural regeneration of the woodland is attributed to extensive livestock farming. For this reason, the economic effects of the livestock activity, whether improvement or degradation of the provision of public products, are not taken into account in the economic accounts of the case-study *dehesa* owners. The livestock farming activity, given the property rights regulated by social contracts (laws and customs), can be orientated toward sustainable forms of management with regard to the natural environment and social interest, through voluntary actions agreed among landowners and public administrations. In this instance, the concerted action could be designed and implemented taking into account the estimates of intermediate products of self-consumed amenity services (ISSnca).

ISSnca estimates of zero value in situations where a net operating margin at basic price equal to or above the normal net operating margin is estimated imply that the AAS application could estimate the exact volume of government compensation (operating subsidies net of taxes linked to production) and avoid either under- or overcompensation to the owner [26].

## 5. Concluding Remarks

Although the objective of this article was not to study in depth the development of the cultural landscape of the Mediterranean silvo-pastoral systems of the case-study *dehesa* farms, we did establish that the rearing of livestock is the "raison d'être" of the savannah like landscape of the *dehesas* with the aim of encouraging the biological productivity of the pasture grazed by the livestock species. This development of the cultural landscape of the *dehesa* bears similarity to that of a consumable inanimate construction, and the restoration of the woodland, degraded through aging as a result of insufficient recruitment of natural regeneration caused by the consumption of pasture, is the pending issue to be resolved. As long as the continuity of livestock rearing on the *dehesa* landscapes is considered of public interest, it is the public policy of landscape conservation and landowner preference for assuming voluntary opportunity costs which must work together in the restoration of woodland of the *Quercus* genus in the *dehesas* and any other Mediterranean silvo-pastoral system.

The official FADN methodology in concordance with the Economic Accounts for Agriculture and Forestry (EEA-EAF) is intended to estimate the net value added of the production of goods from agricultural farms and the national territory, respectively. Neither of these two methodologies estimates the noncommercial private amenity production self-consumed by the nonindustrial owners of the farms [27,28]. This omission is apparent in the estimates of the net operating margins (surpluses) at basic prices under these official methodologies which are frequently noncompetitive or even negative for farms predominantly orientated toward mixed livestock rearing.

The historical persistence of livestock rearing by large nonindustrial *dehesa* owners is not only due to the commercial livestock production margins at basic prices, but also due to the noncommercial intermediate products of private amenity services (ISSnca) associated with these species, which may become the main reason for the continuity of the livestock farming activity.

The AAS methodology applied in this research to the case-study private *dehesas* estimates the ISSnca under the hypothesis of voluntary opportunity cost accepted by the owner, through comparison with the normal net operating margin which would be obtained from the investment of immobilized livestock capital in an alternative nonagricultural commercial asset. The result of incorporating the production of self-consumed private amenity services is that the value added of livestock farming at social prices, measured using the AAS methodology, is 4.2 times what it would be under the official FADN methodology.

The final corollary of this research is that the government should extend the FADN methodology to incorporate the self-consumption of noncommercial production of services in the value added of the livestock farming activity when it is observed that the voluntary opportunity costs of the livestock farmers persist over time.

**Supplementary Materials:** The following are available online at www.mdpi.com/2077-0472/11/3/214/s1: Text S1. Estimation of capital of building and equipment per livestock species; Table S1. Livestock races by species on large privately owned case-study *dehesas* in Andalusia (2010); Table S2. Meat cattle physical indicators of large privately owned case-study *dehesas* in Andalusia (2010); Table S3. Fighting bull physical indicators of large privately owned case-study *dehesas* in Andalusia (2010); Table S4. Sheep physical indicators of large privately owned case-study *dehesas* in Andalusia (2010); Table S5. Goat physical indicators of large privately owned case-study *dehesas* in Andalusia (2010); Table S6. *Montanera* Iberian pig physical indicators of large privately owned case-study *dehesas* in Andalusia (2010); Table S7. *Montanera* Iberian pig grazing season on large privately owned case-study *dehesas* in Andalusia (2010); Table S8. Extensive piglet physical indicators of a large privately owned case-study *dehesa* in Andalusia (2010); Table S9. Horse physical indicators of large privately owned case-study *dehesas* in Andalusia (2010); Table S10. Bee physical indicators of large privately owned case-study *dehesas* in Andalusia (2010).

**Author Contributions:** Conceptualization, P.C.; data curation, P.C., B.M., and A.Á.; formal analysis, P.C., B.M., and A.Á.; funding acquisition, P.C.; methodology, P.C.; project administration, P.C.; supervision, P.C.; visualization, P.C. and B.M.; writing—original draft, P.C., B.M.; writing—review and editing, P.C., B.M., and A.Á. All authors have read and agreed to the published version of the manuscript.

**Funding:** This study was funded by the Agency for Water and Environment of the Regional Government of Andalusia, Contract NET 165602 and the Spanish National Research Council (CSIC), grant number ref. 201810E036.

**Data Availability Statement:** The data presented in this study are available on request from the corresponding author. The data are not publicly available due to ownership rights.

**Acknowledgments:** The authors thank the Agency for Water and Environment of the Regional Government of Andalusia for the financial and field work support for the Renta y Capital de los Montes de Andalucía (RECAMAN) project (Contract NET 165602) and the Valoraciones de Servicios y Activos de Amenidades Privadas de Fincas Silvopastorales (VAMSIL) project of CSIC (ref. 201810E036). We acknowledge the contributions of Alejandro Caparrós, José L. Oviedo, Paola Ovando, Eloy Almazán, Cristina Fernández, Soledad Letón, and other colleagues in the framework of the RECAMAN project to the methods and results presented in this study. We thank Adam B. Collins for helping us to review the English writing.

**Conflicts of Interest:** The authors declare no conflicts of interest.

## Appendix A

**Table A1.** Livestock and game species grazing and supplement consumption of large privately owned *dehesas* in Andalusia (2010: FU/ha).

| Class | Grazing | | | | | | | | | Supplements | Total |
|---|---|---|---|---|---|---|---|---|---|---|---|
| | Grass and Browse | | | Acorn | | | Total | | | | |
| | Commercial | Free | Total | Commercial | Free | Total | Commercial | Free | Total | | |
| Livestock (*n* = 21) | 229.5 | 24.6 | 254.1 | 34.3 | | 34.3 | 263.8 | 24.6 | 288.5 | 386.6 | 675.1 |
| Meat cattle (*n* = 12) | 83.9 | 10.7 | 94.6 | | | | 83.9 | 10.7 | 94.6 | 36.6 | 131.2 |
| Fighting bulls (*n* = 2) | 74.7 | 6.5 | 81.2 | | | | 74.7 | 6.5 | 81.2 | 90.5 | 171.7 |
| Sheep (*n* = 8) | 22.6 | 2.6 | 25.2 | 1.6 | | 1.6 | 24.1 | 2.6 | 26.8 | 21.8 | 48.6 |
| Goats (*n* = 6) | 7.7 | 3.6 | 11.3 | 0.9 | | 0.9 | 8.6 | 3.6 | 12.2 | 53.6 | 65.8 |
| *Montanera* pigs (*n* = 9) | 30.8 | 1.1 | 31.8 | 31.8 | | 31.8 | 62.5 | 1.1 | 63.6 | 142.6 | 206.2 |
| Extensive piglets (*n* = 1) | | | | 0.1 | | 0.1 | 0.1 | | 0.1 | 0.8 | 0.9 |
| Horses (*n* = 8) | 9.9 | 0.2 | 10.0 | | | | 9.9 | 0.2 | 10.0 | 40.6 | 50.7 |
| Hunting (*n* = 21) | 37.8 | 225.3 | 263.1 | 13.5 | 0.6 | 14.0 | 51.3 | 225.9 | 277.2 | 29.1 | 306.3 |
| Red deer (*n* = 11) | 33.9 | 165.6 | 199.5 | 11.6 | 0.1 | 11.6 | 45.4 | 165.7 | 211.1 | 11.8 | 223.0 |
| Wild boar (*n* = 11) | 3.5 | 37.3 | 40.9 | 1.9 | 0.5 | 2.4 | 5.4 | 37.9 | 43.3 | 16.9 | 60.2 |
| Other species (*n* = 16) | 0.4 | 22.3 | 22.7 | 0.0 | | 0.0 | 0.4 | 22.3 | 22.7 | 0.4 | 23.2 |
| Total (*n* = 21) | 267.3 | 249.9 | 517.3 | 47.8 | 0.6 | 48.4 | 315.1 | 250.5 | 565.7 | 415.7 | 981.4 |

*Source*: Own elaboration. *Abbreviations*: *n* is the number of *dehesas* with the presence of this species more than 6 months per year (total = 21); FU is the forage unit. *Notes:* A forage unit refers to the energy content of a kilogram of barley with a humidity content of 14.1% and totals 2723 kcal [18]. The total area of case-study *dehesas* is 15,372 hectares. The average area of case-study *dehesas* is 732 hectares.

**Table A2.** Livestock species labor quantity, price, and value of large privately owned *dehesas* in Andalusia (2010).

| Class | Unit (u) | Employee | | | Self-Employed | | | | Total | | | |
|---|---|---|---|---|---|---|---|---|---|---|---|---|
| | | | | | Without Price | With Price | | | Without Price | With Price | | |
| | | Quantity (h/u) | Price (EUR/h) | Value (EUR/u) | Quantity (h/u) | Quantity (h/u) | Price (EUR/h) | Value (EUR/u) | Quantity (h/u) | Quantity (h/u) | Price (EUR/h) | Value (EUR/u) |
| Meat cattle (*n* = 12) | LU | 17.0 | 8.0 | 136.2 | 2.8 | | | | 2.8 | 17.0 | 8.0 | 136.2 |
| Fighting bulls (*n* = 2) | LU | 13.3 | 7.0 | 92.9 | | | | | | 13.3 | 7.0 | 92.9 |
| Sheep (*n* = 8) | LU | 10.0 | 8.8 | 88.7 | 1.9 | 4.5 | 6.1 | 27.1 | 1.9 | 14.5 | 8.0 | 115.8 |
| Goats (*n* = 6) | LU | 29.4 | 9.1 | 268.4 | 25.2 | | | | 25.2 | 29.4 | 9.1 | 268.4 |
| *Montanera* pigs (*n* = 9) | head [(1)] | 4.8 | 11.7 | 56.2 | 0.4 | | | | 0.4 | 4.8 | 11.7 | 56.2 |
| Extensive piglets (*n* = 1) | head sold | 11.2 | 9.9 | 111.3 | | | | | | 11.2 | 9.9 | 111.3 |
| Horses (*n* = 8) | LU | 43.0 | 7.0 | 302.7 | 0.0 | 0.1 | 8.5 | 1.1 | 0.0 | 43.1 | 7.0 | 303.8 |
| Bees (*n* = 5) | hive | 0.1 | 11.4 | 0.7 | 3.1 | 0.0 | 4.3 | 0.0 | 3.1 | 0.1 | 10.2 | 0.7 |

*Source*: Own elaboration. *Abbreviations*: *n* is the number of *dehesas* with the presence of this species more than 6 months per year (total = 21). *Note:* [(1)] Average number of Iberian pigs in *montanera* per year and *dehesa*. A livestock unit is estimated as a coefficient of the annual energy requirements of an empty Retinta cow with a weight of 450 kg [22]. An LU is equal to an annual requirement of 5171.32 Mcal of metabolizable energy. For meat cattle, sheep, and goats, it is equal to the equivalent LU of adult breeders. For fighting bulls and horses, it is equal to the equivalent LU of the animals older than 1 year. Absolute unit measures are as follows: meat cattle, 988 LU; fighting bulls, 1896 LU; sheep, 465 LU; goats, 241 LU; *montanera* pigs, 2758 heads; extensive piglets, 93 heads; horses, 330 LU; bees, 690 hives. An annual work unit is equivalent to 1826 h worked per year [23]. The total area of case-study *dehesas* is 15,372 hectares. The average area of case-study *dehesas* is 732 hectares.

**Table A3.** Livestock species production account under the AAS for large privately owned *dehesas* in Andalusia (2010: EUR/u).

| Class | Meat Cattle (n = 12) (EUR/LU) | Fighting Bulls (n = 2) (EUR/LU) | Sheep (n = 8) (EUR/LU) | Goats (n = 6) (EUR/LU) | *Montanera* Pigs (n = 9) (EUR/head [1]) | Extensive Piglets (n = 1) (EUR/head sold) | Horses (n = 8) (EUR/LU) | Bees (n = 5) (EUR/hive) |
|---|---|---|---|---|---|---|---|---|
| 1. Total product (TP) | 1101.0 | 834.6 | 645.2 | 1241.4 | 877.7 | 366.3 | 2389.3 | 33.8 |
| 1.1 Intermediate product (IP) | 616.3 | 161.7 | 232.7 | 611.4 | 83.5 | 187.5 | 566.4 | 24.0 |
| Intermediate raw materials (IRM) | | | | | | | | 4.2 |
| Intermediate services (ISS) | 616.3 | 161.7 | 232.7 | 611.4 | 83.5 | 187.5 | 566.4 | 19.8 |
| Compensated (ISSncc) | 279.2 | 77.0 | 162.8 | 162.3 | 0.3 | | | 0.6 |
| Amenity auto-consumed (ISSnca) | 337.1 | 84.7 | 69.8 | 449.1 | 83.2 | 187.5 | 566.4 | 19.2 |
| 1.2 Final product (FP) | 484.7 | 672.9 | 412.5 | 630.0 | 794.2 | 178.8 | 1822.9 | 9.9 |
| Sales (FPs) | 216.5 | 146.8 | 339.2 | 489.8 | 446.1 | 125.7 | 20.0 | 8.8 |
| Gross fixed capital formation (GFCF) | 80.7 | 139.1 | 15.9 | 69.5 | 10.4 | 12.7 | 415.7 | |
| Gross work in progress formation (GWPF) | 187.4 | 378.9 | 56.1 | 69.5 | 336.6 | 21.6 | 1387.2 | |
| Other final product (FPo) | 0.0 | 8.1 | 1.3 | 1.1 | 1.1 | 18.8 | | 1.1 |
| 2. Total cost (TC) | 951.3 | 797.8 | 492.5 | 1162.9 | 844.1 | 354.2 | 2196.3 | 32.3 |
| 2.1. Intermediate consumption (IC) | 724.8 | 691.9 | 320.3 | 766.2 | 774.6 | 224.6 | 1766.6 | 23.1 |
| Raw materials (RM) | 388.4 | 180.3 | 236.3 | 590.3 | 265.6 | 62.1 | 469.2 | 10.8 |
| Bought (RMb) | 315.5 | 152.1 | 196.0 | 549.3 | 216.7 | 59.2 | 432.8 | 6.6 |
| Own (RMo) | 72.9 | 28.2 | 40.2 | 41.0 | 48.8 | 2.9 | 36.4 | 4.2 |
| Services (SS) | 88.1 | 40.1 | 34.6 | 46.7 | 29.7 | 10.4 | 159.7 | 12.2 |
| Bought (SSb) | 88.1 | 40.1 | 34.6 | 46.7 | 29.7 | 10.4 | 159.7 | 12.2 |
| Work in progress used (WPu) | 248.2 | 471.6 | 49.4 | 129.1 | 479.3 | 152.1 | 1137.7 | |
| 2.2 Labor cost (LC) | 136.2 | 92.9 | 115.8 | 268.4 | 56.2 | 111.3 | 303.8 | 0.7 |
| Employee (LCe) | 136.2 | 92.9 | 88.7 | 268.4 | 56.2 | 111.3 | 302.7 | 0.7 |
| Self-employed (LCse) | 0.0 | | 27.1 | 0.0 | 0.0 | | 1.1 | 0.0 |
| 2.3 Consumption of fixed capital (CFC) | 90.3 | 13.0 | 56.4 | 128.3 | 13.3 | 18.3 | 125.8 | 8.5 |
| 3. Net operating margin (NOM) | 149.7 | 36.8 | 152.7 | 78.5 | 33.6 | 12.1 | 193.0 | 1.5 |
| 4. Gross valued added (GVA) | 376.2 | 142.7 | 324.9 | 475.2 | 103.1 | 141.7 | 622.7 | 10.8 |
| 5. Net valued added (NVA) | 285.9 | 129.7 | 268.5 | 346.9 | 89.8 | 123.4 | 496.8 | 2.2 |

*Source*: Own elaboration. *Abbreviations*: *n* is the number of *dehesas* with the presence of this species more than 6 months per year (total = 21); LU is the livestock unit. *Note:* [1] Average number of Iberian pigs in *montanera* per year and *dehesa*. A livestock unit is estimated as a coefficient of the annual energy requirements of an empty Retinta cow with a weight of 450 kg [22]. An LU is equal to an annual requirement of 5171.32 Mcal of metabolizable energy. For meat cattle, sheep, and goats, it is equal to the equivalent LU of adult breeders. For fighting bulls and horses, it is equal to the equivalent LU of the animals older than 1 year. Absolute unit measures are as follows: meat cattle, 988 LU; fighting bulls, 1896 LU; sheep, 465 LU; goats, 241 LU; *montanera* pigs, 2758 heads; extensive piglets, 93 heads; horses, 330 LU; bees, 690 hives. An annual work unit is equivalent to 1826 h worked per year [23]. The total area of case-study *dehesas* is 15,372 hectares. The average area of case-study *dehesas* is 732 hectares.

**Table A4.** Livestock species capital balance under the AAS of large privately owned *dehesas* in Andalusia (2010: EUR/u).

| Class | Unit (u) | 1. Opening Capital (Co) | 2. Capital Entries | | | | 3. Capital Withdrawals | | | | | | 4. Revaluation (Cr) | 5. Closing Capital (Cc) |
|---|---|---|---|---|---|---|---|---|---|---|---|---|---|---|
| | | | 2.1 Bought (Ceb) | 2.2 Own (Ceo) | 2.3 Others (Ceot) | 2.4 Total (Ce) | 3.1 Used (Cwu) | 3.2 Sales (Cws) | 3.2 Destructions (Cwd) | 3.3. Reclassifications (Cwrc) | 3.4 Others (Cwo) | 3.5 Total (Cw) | | |
| 1. Capital (C = WP + FC) | | | | | | | | | | | | | | |
| Meat cattle (*n* = 12) | EUR/LU | 4079.7 | 82.3 | 230.5 | 3.0 | 315.8 | 248.2 | 70.2 | 41.8 | | | 360.2 | −170.9 | 3864.4 |
| Fighting bulls (*n* = 2) | EUR/LU | 1163.9 | | 516.1 | 1.9 | 518.0 | 471.6 | 15.8 | 21.9 | | 87.6 | 596.9 | −23.3 | 1061.7 |
| Sheep (*n* = 8) | EUR/LU | 1990.0 | 0.3 | 72.0 | 0.0 | 72.3 | 49.4 | 16.7 | 22.6 | | 0.5 | 89.2 | −96.2 | 1876.9 |
| Goats (*n* = 6) | EUR/LU | 2294.6 | | 139.1 | | 139.1 | 129.1 | 12.5 | 24.9 | | | 166.5 | −299.0 | 1968.2 |
| *Montanera* pigs (*n* = 9) | EUR/head [(1)] | 603.4 | 81.0 | 229.4 | 44.6 | 355.0 | 479.3 | | 2.8 | | | 482.1 | −11.2 | 465.0 |
| Extensive piglets (*n* = 1) | EUR/head sold | 382.6 | | 34.3 | | 34.3 | 152.1 | 11.1 | | | | 163.2 | −4.3 | 249.4 |
| Horses (*n* = 8) | EUR/LU | 5873.5 | 24.7 | 1760.4 | 284.9 | 2070.0 | 1137.7 | 463.4 | | | 71.9 | 1673.0 | −48.4 | 6222.0 |
| Bees (*n* = 5) | EUR/hive | 43.7 | | | | | | | | | | | −3.3 | 40.4 |
| 2. Work in progress (WP) | | | | | | | | | | | | | | |
| Meat cattle (*n* = 12) | EUR/LU | 248.2 | 34.5 | 149.9 | 3.0 | 187.4 | 248.2 | | | | | 248.2 | 0.0 | 187.4 |
| Fighting bulls (*n* = 2) | EUR/LU | 471.6 | | 377.0 | 1.9 | 378.9 | 471.6 | | | | | 471.6 | 0.0 | 378.9 |
| Sheep (*n* = 8) | EUR/LU | 49.4 | | 56.1 | | 56.1 | 49.4 | | | | | 49.4 | 0.0 | 56.1 |
| Goats (*n* = 6) | EUR/LU | 129.1 | | 69.5 | | 69.5 | 129.1 | | | | | 129.1 | 0.0 | 69.5 |
| *Montanera* pigs (*n* = 9) | EUR/head | 479.3 | 78.1 | 219.0 | 39.4 | 336.6 | 479.3 | | | | | 479.3 | 0.0 | 336.6 |
| Extensive piglets (*n* = 1) | EUR/head sold | 152.1 | | 21.6 | | 21.6 | 152.1 | | | | | 152.1 | 0.0 | 21.6 |
| Horses (*n* = 8) | EUR/LU | 1137.7 | | 1344.7 | 42.5 | 1387.2 | 1137.7 | | | | | 1137.7 | 0.0 | 1387.2 |
| Bees (*n* = 5) | EUR/hive | | | | | | | | | | | | | |
| 3. Fixed capital (FC = FCa + FCi) | | | | | | | | | | | | | | |
| Meat cattle (*n* = 12) | EUR/LU | 3831.5 | 47.7 | 80.7 | | 128.4 | | 70.2 | 41.8 | | | 112.0 | −170.9 | 3677.0 |
| Fighting bulls (*n* = 2) | EUR/LU | 692.3 | | 139.1 | | 139.1 | | 15.8 | 21.9 | | 87.6 | 125.3 | −23.3 | 682.8 |
| Sheep (*n* = 8) | EUR/LU | 1940.5 | 0.3 | 15.9 | 0.0 | 16.2 | | 16.7 | 22.6 | | 0.5 | 39.8 | −96.2 | 1820.8 |
| Goats (*n* = 6) | EUR/LU | 2165.5 | | 69.5 | | 69.5 | | 12.5 | 24.9 | | | 37.4 | −299.0 | 1898.6 |
| *Montanera* pigs (*n* = 9) | EUR/head | 124.1 | 2.9 | 10.4 | 5.1 | 18.4 | | | 2.8 | | | 2.8 | −11.2 | 128.4 |
| Extensive piglets (*n* = 1) | EUR/head sold | 230.5 | | 12.7 | | 12.7 | | 11.1 | | | | 11.1 | −4.3 | 227.8 |
| Horses (*n* = 8) | EUR/LU | 4735.8 | 24.7 | 415.7 | 242.4 | 682.8 | | 463.4 | | | 71.9 | 535.3 | −48.4 | 4834.8 |
| Bees (*n* = 5) | EUR/hive | 43.7 | | | | | | | | | | | −3.3 | 40.4 |
| 3.1 Alive (FCa) | | | | | | | | | | | | | | |
| Meat cattle (*n* = 12) | EUR/LU | 897.7 | 37.8 | 80.7 | | 118.4 | | 70.2 | 37.8 | | | 108.0 | 5.5 | 913.7 |
| Fighting bulls (*n* = 2) | EUR/LU | 460.5 | | 139.1 | | 139.1 | | 15.8 | 21.9 | | 87.6 | 125.3 | 0.6 | 474.9 |
| Sheep (*n* = 8) | EUR/LU | 379.7 | 0.3 | 15.9 | 0.0 | 16.2 | | 16.7 | 22.6 | | 0.5 | 39.8 | −24.3 | 331.9 |
| Goats (*n* = 6) | EUR/LU | 245.9 | | 69.5 | | 69.5 | | 12.5 | 15.3 | | | 27.8 | −3.0 | 284.7 |
| *Montanera* pigs (*n* = 9) | EUR/head | 11.7 | | 10.4 | 5.1 | 15.5 | | | 2.7 | | | 2.7 | 0.0 | 24.5 |

| | | | | | | | | | | | | |
|---|---|---|---|---|---|---|---|---|---|---|---|---|
| Extensive piglets (*n* = 1) | EUR/head sold | 29.0 | | 12.7 | | 12.7 | 11.1 | | | 11.1 | 2.6 | 33.2 |
| Horses (*n* = 8) | EUR/LU | 3686.3 | 24.0 | 415.7 | 242.4 | 682.1 | 463.4 | | 71.9 | 535.3 | 1.8 | 3834.9 |
| Bees (*n* = 5) | EUR/hive | | | | | | | | | | | |
| 3.2 Inanimate (FCi) | | | | | | | | | | | | |
| Meat cattle (*n* = 12) | EUR/LU | 2933.7 | 10.0 | | | 10.0 | 4.0 | | | 4.0 | −176.4 | 2763.3 |
| Fighting bulls (*n* = 2) | EUR/LU | 231.8 | | | | | | | | | −23.9 | 207.9 |
| Sheep (*n* = 8) | EUR/LU | 1560.9 | | | | | | | | | −71.9 | 1488.9 |
| Goats (*n* = 6) | EUR/LU | 1919.5 | | | | | 9.6 | | | 9.6 | −296.0 | 1614.0 |
| *Montanera* pigs (*n* = 9) | EUR/head | 112.5 | 2.9 | | | 2.9 | 0.1 | | | 0.1 | −11.3 | 104.0 |
| Extensive piglets (*n* = 1) | EUR/head sold | 201.5 | | | | | | | | | −6.9 | 194.6 |
| Horses (*n* = 8) | EUR/LU | 1049.5 | 0.7 | | | 0.7 | | | | | −50.2 | 1000.0 |
| Bees (*n* = 5) | EUR/hive | 43.7 | | | | | | | | | −3.3 | 40.4 |

*Source*: Own elaboration. *Abbreviations*: *n* is the number of *dehesas* with the presence of this species more than 6 months per year (total = 21); LU is the livestock unit. *Note:* [1] Average number of Iberian pigs in *montanera* per year and *dehesa*. A livestock unit is estimated as a coefficient of the annual energy requirements of an empty Retinta cow with a weight of 450 kg [22]. An LU is equal to an annual requirement of 5171.32 Mcal of metabolizable energy. For meat cattle, sheep, and goats, it is equal to the equivalent LU of adult breeders. For fighting bulls and horses, it is equal to the equivalent LU of the animals older than 1 year. Absolute unit measures are as follows: meat cattle, 988 LU; fighting bulls, 1896 LU; sheep, 465 LU; goats, 241 LU; *montanera* pigs, 2758 heads; extensive piglets, 93 heads; horses, 330 LU; bees, 690 hives. An annual work unit is equivalent to 1826 h worked per year [23]. The total area of case-study *dehesas* is 15,372 hectares. The average area of case-study *dehesas* is 732 hectares.

**Table A5.** Livestock species operating cash flow of large privately owned *dehesas* in Andalusia (2010: EUR/u).

| Class | Meat Cattle (n = 12) (EUR/LU) | Fighting Bulls (n = 2) (EUR/LU) | Sheep (n = 8) (EUR/LU) | Goats (n = 6) (EUR/LU) | Montanera Pigs (n = 9) (EUR/head [1]) | Extensive Piglets (n = 1) (EUR/head sold) | Horses (n = 8) (EUR/LU) | Bees (n = 5) (EUR/hive) |
|---|---|---|---|---|---|---|---|---|
| 1. Revenue | 566.0 | 247.7 | 520.0 | 665.8 | 447.5 | 155.6 | 483.4 | 10.4 |
| 1.1 Sales | 286.8 | 162.5 | 355.9 | 502.4 | 446.1 | 136.8 | 483.4 | 8.8 |
| 1.2 Auto-consumption | 0.0 | 8.1 | 1.3 | 1.1 | 1.1 | 18.8 | | 1.1 |
| 1.3 Compensations | 279.2 | 77.0 | 162.8 | 162.3 | 0.3 | 0.0 | | 0.6 |
| 2. Expenditure | 712.5 | 298.0 | 376.1 | 992.8 | 397.0 | 199.1 | 1045.7 | 28.1 |
| 2.1 Bought raw material | 315.5 | 152.1 | 196.0 | 549.3 | 216.7 | 59.2 | 432.8 | 6.6 |
| 2.2 Bought services | 88.1 | 40.1 | 34.6 | 46.7 | 29.7 | 10.4 | 159.7 | 12.2 |
| 2.3 Bought livestock | 82.3 | 0.0 | 0.3 | 0.0 | 81.0 | 0.0 | 24.7 | 0.0 |
| 2.4 Employee labor cost | 136.2 | 92.9 | 88.7 | 268.4 | 56.2 | 111.3 | 302.7 | 0.7 |
| 2.5 Consumption of fixed capital | 90.3 | 13.0 | 56.4 | 128.3 | 13.3 | 18.3 | 125.8 | 8.5 |
| 3. Operating cash flow | −146.5 | −50.3 | 144.0 | −327.0 | 50.5 | −43.5 | −562.3 | −17.7 |

*Source*: Own elaboration. *Abbreviations*: *n* is the number of *dehesas* with the presence of this species more than 6 months per year (total = 21); LU is the livestock unit. *Note:* [1] Average number of Iberian pigs in *montanera* per year and *dehesa*. A livestock unit is estimated as a coefficient of the annual energy requirements of an empty Retinta cow with a weight of 450 kg [22]. An LU is equal to an annual requirement of 5171.32 Mcal of metabolizable energy. For meat cattle, sheep, and goats, it is equal to the equivalent LU of adult breeders. For fighting bulls and horses, it is equal to the equivalent LU of the animals older than 1 year. Absolute unit measures are as follows: meat cattle, 988 LU; fighting bulls, 1896 LU; sheep, 465 LU; goats, 241 LU; *montanera* pigs, 2758 heads; extensive piglets, 93 heads; horses, 330 LU; bees, 690 hives. An annual work unit is equivalent to 1826 h worked per year [23]. The total area of case-study *dehesas* is 15,372 hectares. The average area of case-study *dehesas* is 732 hectares.

**Table A6.** Livestock species and activity production account under the AAS of large privately owned *dehesas* in Andalusia (2010: EUR/ha).

| Class | Meat Cattle (n = 12) | Fighting Bulls (n = 2) | Sheep (n = 8) | Goats (n = 6) | Montanera Pigs (n = 9) | Extensive Piglets (n = 1) | Horses (n = 8) | Bees (n = 5) | Livestock (n = 21) |
|---|---|---|---|---|---|---|---|---|---|
| 1. Total product (TP) | 70.7 | 102.9 | 19.5 | 19.5 | 157.5 | 2.2 | 51.4 | 1.5 | 425.3 |
| 1.1 Intermediate product (IP) | 39.6 | 19.9 | 7.0 | 9.6 | 15.0 | 1.1 | 12.2 | 1.1 | 105.6 |
| Intermediate raw materials (IRM) | | | | | | | | 0.2 | 0.2 |
| Intermediate services (ISS) | 39.6 | 19.9 | 7.0 | 9.6 | 15.0 | 1.1 | 12.2 | 0.9 | 105.4 |
| Compensated (ISSncc) | 17.9 | 9.5 | 4.9 | 2.6 | 0.1 | | | 0.0 | 35.0 |
| Amenity auto-consumed (ISSnca) | 21.7 | 10.4 | 2.1 | 7.1 | 14.9 | 1.1 | 12.2 | 0.9 | 70.4 |
| 1.2 Final product (FP) | 31.1 | 83.0 | 12.5 | 9.9 | 142.5 | 1.1 | 39.2 | 0.4 | 319.7 |
| Sales (FPs) | 13.9 | 18.1 | 10.3 | 7.7 | 80.0 | 0.8 | 0.4 | 0.4 | 131.6 |
| Gross fixed capital formation (GFCF) | 5.2 | 17.2 | 0.5 | 1.1 | 1.9 | 0.1 | 8.9 | | 34.8 |
| Gross work in progress formation (GWPF) | 12.0 | 46.7 | 1.7 | 1.1 | 60.4 | 0.1 | 29.8 | | 151.9 |
| Other final product (FPo) | 0.0 | 1.0 | 0.0 | 0.0 | 0.2 | 0.1 | | 0.0 | 1.4 |
| 2. Total cost (TC) | 61.1 | 98.4 | 14.9 | 18.3 | 151.4 | 2.1 | 47.2 | 1.5 | 394.9 |
| 2.1. Intermediate consumption (IC) | 46.6 | 85.3 | 9.7 | 12.0 | 139.0 | 1.4 | 38.0 | 1.0 | 333.0 |
| Raw materials (RM) | 25.0 | 22.2 | 7.1 | 9.3 | 47.7 | 0.4 | 10.1 | 0.5 | 122.2 |
| Bought (RMb) | 20.3 | 18.8 | 5.9 | 8.6 | 38.9 | 0.4 | 9.3 | 0.3 | 102.4 |
| Own (RMo) | 4.7 | 3.5 | 1.2 | 0.6 | 8.8 | 0.0 | 0.8 | 0.2 | 19.8 |
| Services (SS) | 5.7 | 4.9 | 1.0 | 0.7 | 5.3 | 0.1 | 3.4 | 0.5 | 21.8 |
| Bought (SSb) | 5.7 | 4.9 | 1.0 | 0.7 | 5.3 | 0.1 | 3.4 | 0.5 | 21.8 |
| Work in progress used (WPu) | 16.0 | 58.2 | 1.5 | 2.0 | 86.0 | 0.9 | 24.5 | | 189.0 |
| 2.2 Labor cost (LC) | 8.8 | 11.5 | 3.5 | 4.2 | 10.1 | 0.7 | 6.5 | 0.0 | 45.2 |
| Employee (LCe) | 8.8 | 11.5 | 2.7 | 4.2 | 10.1 | 0.7 | 6.5 | 0.0 | 44.4 |
| Self-employed (LCse) | | | 0.8 | | | | | 0.0 | 0.8 |
| 2.3 Consumption of fixed capital (CFC) | 5.8 | 1.6 | 1.7 | 2.0 | 2.4 | 0.1 | 2.7 | 0.4 | 16.7 |
| 3. Net operating margin (NOM) | 9.6 | 4.5 | 4.6 | 1.2 | 6.0 | 0.1 | 4.1 | 0.1 | 30.3 |
| 4. Gross valued added (GVA) | 24.2 | 17.6 | 9.8 | 7.5 | 18.5 | 0.9 | 13.4 | 0.5 | 92.3 |
| 5. Net valued added (NVA) | 18.4 | 16.0 | 8.1 | 5.4 | 16.1 | 0.7 | 10.7 | 0.1 | 75.6 |

*Source*: Own elaboration. *Abbreviations*: *n* is the number of *dehesas* with the presence of this species more than 6 months per year. The total area of case-study *dehesas* is 15,372 hectares. The average area of case-study *dehesas* is 732 hectares.

**Table 7.** Livestock species capital balance under the AAS of large privately owned *dehesas* in Andalusia (2010: EUR/ha).

| Class | 1. Opening Capital | 2. Capital Entries | | | | 3. Capital Withdrawals | | | | | | 4. Revaluation | 5. Closing Capital |
|---|---|---|---|---|---|---|---|---|---|---|---|---|---|
| | | 2.1 Bought | 2.2 Own | 2.3 Others | 2.4 Total | 3.1 Used | 3.2 Sales | 3.2 Destructions | 3.3. Reclassifications | 3.4 Others | 3.5 Total | | |
| | (Co) | (Ceb) | (Ceo) | (Ceot) | (Ce) | (Cwu) | (Cws) | (Cwd) | (Cwrc) | (Cwo) | (Cw) | (Cr) | (Cc) |
| 1. Capital (C = WP + FC) | 740.7 | 20.4 | 162.0 | 14.5 | 196.9 | 189.0 | 17.2 | 7.0 | | 12.4 | 225.5 | −24.7 | 687.4 |
| Meat cattle (*n*= 12) | 262.1 | 5.3 | 14.8 | 0.2 | 20.3 | 16.0 | 4.5 | 2.7 | | | 23.1 | −11.0 | 248.3 |
| Fighting bulls (*n* = 2) | 143.5 | | 63.7 | 0.2 | 63.9 | 58.2 | 1.9 | 2.7 | | 10.8 | 73.6 | −2.9 | 130.9 |
| Sheep (*n* = 8) | 60.2 | 0.0 | 2.2 | 0.0 | 2.2 | 1.5 | 0.5 | 0.7 | | 0.0 | 2.7 | −2.9 | 56.8 |
| Goats (*n* = 6) | 36.0 | | 2.2 | | 2.2 | 2.0 | 0.2 | 0.4 | | | 2.6 | −4.7 | 30.9 |
| *Montanera* pigs (*n* = 9) | 108.3 | 14.5 | 41.2 | 8.0 | 63.7 | 86.0 | | 0.5 | | | 86.5 | −2.0 | 83.4 |
| Extensive piglets (*n* = 1) | 2.3 | | 0.2 | | 0.2 | 0.9 | 0.1 | | | | 1.0 | 0.0 | 1.5 |
| Horses (*n* = 8) | 126.3 | 0.5 | 37.8 | 6.1 | 44.5 | 24.5 | 10.0 | | | 1.5 | 36.0 | −1.0 | 133.8 |
| Bees (*n* = 5) | 2.0 | | | | | | | | | | | −0.1 | 1.8 |
| 2. Work in progress (WP) | 189.0 | 16.2 | 127.3 | 8.4 | 151.9 | 189.0 | | | | | 189.0 | 0.0 | 151.9 |
| Meat cattle (*n* = 12) | 16.0 | 2.2 | 9.6 | 0.2 | 12.0 | 16.0 | | | | | 16.0 | 0.0 | 12.0 |
| Fighting bulls (*n* = 2) | 58.2 | | 46.5 | 0.2 | 46.7 | 58.2 | | | | | 58.2 | 0.0 | 46.7 |
| Sheep (*n* = 8) | 1.5 | | 1.7 | | 1.7 | 1.5 | | | | | 1.5 | 0.0 | 1.7 |
| Goats (*n* = 6) | 2.0 | | 1.1 | | 1.1 | 2.0 | | | | | 2.0 | 0.0 | 1.1 |
| *Montanera* pigs (*n* = 9) | 86.0 | 14.0 | 39.3 | 7.1 | 60.4 | 86.0 | | | | | 86.0 | 0.0 | 60.4 |
| Extensive piglets (*n* = 1) | 0.9 | | 0.1 | | 0.1 | 0.9 | | | | | 0.9 | 0.0 | 0.1 |
| Horses (*n* = 8) | 24.5 | | 28.9 | 0.9 | 29.8 | 24.5 | | | | | 24.5 | 0.0 | 29.8 |
| Bees (*n* = 5) | | | | | | | | | | | | | |
| 3. Fixed capital (FC = FCa + FCi) | 551.7 | 4.1 | 34.8 | 6.1 | 45.0 | | 17.2 | 7.0 | | 12.4 | 36.5 | −24.7 | 535.5 |
| Meat cattle (*n* = 12) | 246.2 | 3.1 | 5.2 | | 8.3 | | 4.5 | 2.7 | | | 7.2 | −11.0 | 236.3 |
| Fighting bulls (*n* = 2) | 85.4 | | 17.2 | | 17.2 | | 1.9 | 2.7 | | 10.8 | 15.5 | −2.9 | 84.2 |
| Sheep (*n* = 8) | 58.7 | 0.0 | 0.5 | 0.0 | 0.5 | | 0.5 | 0.7 | | 0.0 | 1.2 | −2.9 | 55.1 |
| Goats (*n* = 6) | 34.0 | | 1.1 | | 1.1 | | 0.2 | 0.4 | | | 0.6 | −4.7 | 29.8 |
| *Montanera* pigs (*n* = 9) | 22.3 | 0.5 | 1.9 | 0.9 | 3.3 | | | 0.5 | | | 0.5 | −2.0 | 23.0 |
| Extensive piglets (*n* = 1) | 1.4 | | 0.1 | | 0.1 | 0.1 | | | | | 0.1 | 0.0 | 1.4 |
| Horses (*n* = 8) | 101.8 | 0.5 | 8.9 | 5.2 | 14.7 | | 10.0 | | | 1.5 | 11.5 | −1.0 | 103.9 |
| Bees (*n* = 5) | 2.0 | | | | | | | | | | | −0.1 | 1.8 |
| 3.1 Alive (FCa) | 211.3 | 3.0 | 34.8 | 6.1 | 43.9 | | 17.2 | 6.5 | | 12.4 | 36.1 | −0.3 | 218.8 |
| Meat cattle (*n* = 12) | 57.7 | 2.4 | 5.2 | | 7.6 | | 4.5 | 2.4 | | | 6.9 | 0.4 | 58.7 |
| Fighting bulls (*n* = 2) | 56.8 | | 17.2 | | 17.2 | | 1.9 | 2.7 | | 10.8 | 15.5 | 0.1 | 58.6 |
| Sheep (*n* = 8) | 11.5 | 0.0 | 0.5 | 0.0 | 0.5 | | 0.5 | 0.7 | | 0.0 | 1.2 | −0.7 | 10.0 |
| Goats (*n* = 6) | 3.9 | | 1.1 | | 1.1 | | 0.2 | 0.2 | | | 0.4 | 0.0 | 4.5 |
| *Montanera* pigs (*n* = 9) | 2.1 | | 1.9 | 0.9 | 2.8 | | | 0.5 | | | 0.5 | 0.0 | 4.4 |
| Extensive piglets (*n* = 1) | 0.2 | | 0.1 | | 0.1 | 0.1 | | | | | 0.1 | 0.0 | 0.2 |
| Horses (*n* = 8) | 79.3 | 0.5 | 8.9 | 5.2 | 14.7 | | 10.0 | | | 1.5 | 11.5 | 0.0 | 82.4 |
| Bees (*n* = 5) | | | | | | | | | | | | | |
| 3.2 Inanimate (FCi) | 340.4 | 1.2 | | | 1.2 | | | 0.4 | | | 0.4 | −24.4 | 316.7 |
| Meat cattle (*n* = 12) | 188.5 | 0.6 | | | 0.6 | | | 0.3 | | | 0.3 | −11.3 | 177.6 |
| Fighting bulls (*n* = 2) | 28.6 | | | | | | | | | | | −2.9 | 25.6 |
| Sheep (*n* = 8) | 47.2 | | | | | | | | | | | −2.2 | 45.0 |
| Goats (*n* = 6) | 30.2 | | | | | | | 0.1 | | | 0.1 | −4.6 | 25.4 |
| *Montanera* pigs (*n* = 9) | 20.2 | 0.5 | | | 0.5 | | | 0.0 | | | 0.0 | −2.0 | 18.7 |
| Extensive piglets (*n* = 1) | 1.2 | | | | | | | | | | | 0.0 | 1.2 |
| Horses (*n* = 8) | 22.6 | 0.0 | | | 0.0 | | | | | | | −1.1 | 21.5 |
| Bees (*n* = 5) | 2.0 | | | | | | | | | | | −0.1 | 1.8 |

*Source*: Own elaboration. *Abbreviations*: *n* is the number of *dehesas* with the presence of this species more than 6 months per year (total = 21). The total area of case-study *dehesas* is 15,372 hectares. The average area of case-study *dehesas* is 732 hectares.

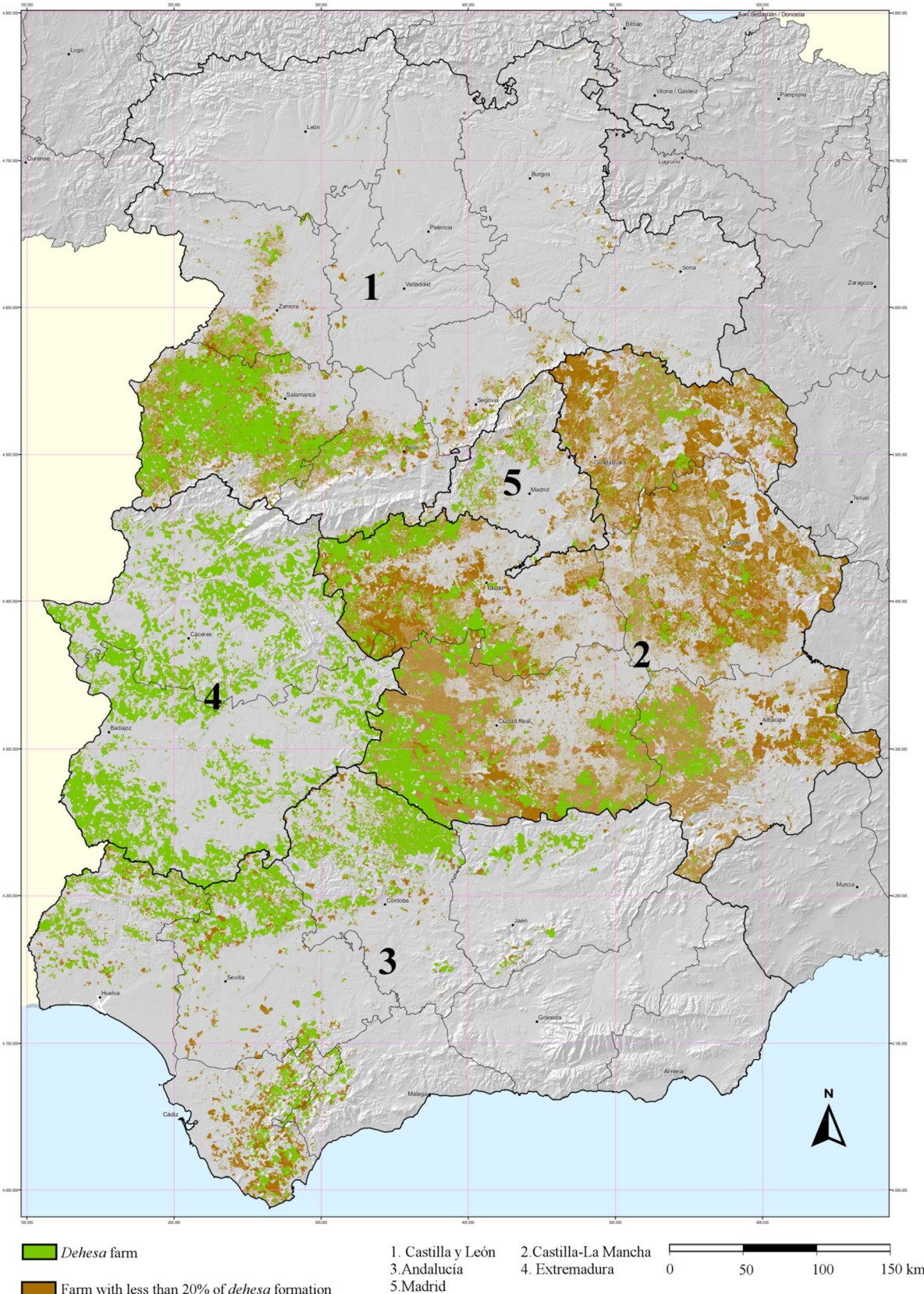

**Figure A1.** *Dehesa* farms in five regions in the west, center, and south of Spain. *Source:* Adapted from [29], Map 3, p. 21.

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
