# Peer review of "Pasture-Based Livestock Economics under Joint Production of Commodities and Private Amenity Self-Consumption: Testing in Large Nonindustrial Privately Owned Dehesa Case Studies in Andalusia, Spain"

_agriculture, doi:10.3390/agriculture11030214_

Round 1
Reviewer 1 Report
This economical study have a very representative experimental area, and with different types of livestock. The author’s results are of interest with regard to the design and application of official economic financial statement at farm scale. The topic of this work is interesting, but there are some points that need to revise:
- In keywords, correct the word compeennsations by compensations.
- Lines 55-70 authors presents their objectives and they are in the middle of the introduction. It is better to explain the objectives at the end of the introduction, before the line 101 where the authors presented the organization of this research.
- In line 116 authors informed that study zone are in the five regions in the west, centre and south of Spain ([4]: Fig. 1, p. 2), they refereed to image and tables in posterior investigation [4]. A representative figure of the different study areas in the present manuscript can make easier to the reader to have information of the study area.
- Agroforestry Accounting System is cited for the first time in the abstract without symbol, and in the line, 71 authors cited AAS. Cite the symbol when we cited for first time the word.
- In the section where the authors explain the strengths and weaknesses of AAS, it is better to put a table with advantages and disadvantages of the system.
- The same, it is better if the authors present a table at the end, summarizing the comparison of the financial calculation systems used for the evaluation of the livestock farming sector (The official Farm Accounting Data Network (FADN) and ASS). This comparative table will help the reader to visualize quickly the differences between the systems and discriminate, which is the best for each type of farm and condition.
Author Response
Manuscript ID: agriculture-1127861
Type of manuscript: Article
Title: Pasture-based livestock economics under joint production of commodities and private amenity self-consumption: testing in large non-industrial privately-owned dehesa case studies in Andalusia-Spain
REVIEWER 1:
Comments and Suggestions for Authors
[Question 1].This economical study have a very representative experimental area, and with different types of livestock. The author’s results are of interest with regard to the design and application of official economic financial statement at farm scale. The topic of this work is interesting, but there are some points that need to revise:
[Authors answer 1]. Thank you for recognizing our effort to test the modelling of the economic rationales for the livestock species production systems based on natural grazing in large private dehesa farms belonging to non-industrial owners of the land and livestock. We are grateful for underlining the necessity to improve this study.
[Question 2]. In keywords, correct the word compeennsations by compensations..
[Authors answer 2]. We have corrected it (line 31).
[Question 3]. Lines 55-70 authors presents their objectives and they are in the middle of the introduction. It is better to explain the objectives at the end of the introduction, before the line 101 where the authors presented the organization of this research..
[Authors answer 3]. Suggestions incorporated. We have moved the objectives to lines 86-100.
[Question 4]. In line 116 authors informed that study zone are in the five regions in the west, centre and south of Spain ([4]: Fig. 1, p. 2), they refereed to image and tables in posterior investigation [4]. A representative figure of the different study areas in the present manuscript can make easier to the reader to have information of the study area.
[Authors answer 4].We have included the new suggested figure (Figure A1) in appendix A (line 1225) and replaced the reference (line 131) for the incorporated one in the map footnote (adapted from [29]).
[Question 5]. Agroforestry Accounting System is cited for the first time in the abstract without symbol, and in the line, 71 authors cited AAS. Cite the symbol when we cited for first time the word.
[Authors answer 5]. Corrected in lines 55-56.
[Question 6] In the section where the authors explain the strengths and weaknesses of AAS, it is better to put a table with advantages and disadvantages of the system.
[Authors answer 6]. We have incorporated a summary table with the main strengths and weakness of AAS (Table 10, line 1018), that are described in detail throughout section 4.2.
[Question 7] The same, it is better if the authors present a table at the end, summarizing the comparison of the financial calculation systems used for the evaluation of the livestock farming sector (The official Farm Accounting Data Network (FADN) and ASS). This comparative table will help the reader to visualize quickly the differences between the systems and discriminate, which is the best for each type of farm and condition.
[Authors answer 7]. We have moved the original Table A8 to the main text (section 3.2.2, Table 9, line 932) in order to help the reader with a quicker comparison of the net value added under the Farm Accounting Data Network (FADN) and the Agroforestry Accounting System (ASS).

Reviewer 2 Report
1. For a rigorous scientific approach, the author must present the research methodology very clear (within the chapter 2 “Materials and methods”). In order to do so, I recommend the author to present, in the beginning of this section which are the main steps / phases of the method (only from theoretical point of view); for each step, a brief description is required, to justify the inclusion of the step in the research method. Just as an example, “Step 1 – Choosing the proper model for analysis” (what theoretical model was used ? which are the main reasons for choosing the model? etc.); “Step 2 Identify the main variables for the analysis” (which are the main variables?) etc.
2. The literature review (and the “References” section) must be improved by adding at least 10 - 15 new titles in the field (to achieve this, I recommend the authors to consult two world-recognized scientific databases - Web of Science and Scopus).
3. Every table or figure in text must present the source. If the table or the figure is the result of the research itself, then the author should mention like this (or similar): “Author’s own conception / calculation, based on XYZ software”.
4. Try to avoid so many bullets & numbers, at the 4th degree (for instance, 2.2.5 is enough, but 2.2.5.1. and 2.2.5.2. is too much). It is a little bit difficult for the reader to understand the main stream of the research.
5. In order to improve the quality of the research paper, the author must present the main findings of the research from multiple angles – from the government (in general, policy makers) perspective, from the local community / individual point of view. Even if the author presents in details the main results from the economic perspective (subchapter 4.1.), it would be very interesting to identify some other perspectives (e.g. social or environmental points of view).
Author Response
Manuscript ID: agriculture-1127861
Type of manuscript: Article
Title: Pasture-based livestock economics under joint production of commodities and private amenity self-consumption: testing in large non-industrial privately-owned dehesa case studies in Andalusia-Spain
REVIEWER 2:
Comments and Suggestions for Authors
[Question 1]. For a rigorous scientific approach, the author must present the research methodology very clear (within the chapter 2 “Materials and methods”). In order to do so, I recommend the author to present, in the beginning of this section which are the main steps / phases of the method (only from theoretical point of view); for each step, a brief description is required, to justify the inclusion of the step in the research method. Just as an example, “Step 1 – Choosing the proper model for analysis” (what theoretical model was used ? which are the main reasons for choosing the model? etc.); “Step 2 Identify the main variables for the analysis” (which are the main variables?) etc.
[Authors answer 1]. We are grateful for help us to improve this study. We have included introductory paragraphs at the beginning of the section 2 according with this recommendation (lines 110-123).
[Question 2]. The literature review (and the “References” section) must be improved by adding at least 10 - 15 new titles in the field (to achieve this, I recommend the authors to consult two world-recognized scientific databases - Web of Science and Scopus).
[Authors answer 2]. Our research applies production and capital accounts to each livestock species and to the other species that are reared on the individual farm in the case study. Unfortunately, we have not found in the literature other references to economic accounts of livestock species grazing on farms that refer to the contribution of livestock species to the farm economy as a whole.
[Question 3]. Every table or figure in text must present the source. If the table or the figure is the result of the research itself, then the author should mention like this (or similar): “Author’s own conception / calculation, based on XYZ software”.
[Authors answer 3]. We have incorporated a footnote on every table and figure with the source of each one.
[Question 4]. Try to avoid so many bullets & numbers, at the 4th degree (for instance, 2.2.5 is enough, but 2.2.5.1. and 2.2.5.2. is too much). It is a little bit difficult for the reader to understand the main stream of the research
[Authors answer 4]. We have eliminated the 4th degree in order to help the reader to follow the concepts presented on the research. The sub-sections of section 2.2 have been renumbered.
[Question 5]. In order to improve the quality of the research paper, the author must present the main findings of the research from multiple angles – from the government (in general, policy makers) perspective, from the local community / individual point of view. Even if the author presents in details the main results from the economic perspective (subchapter 4.1.), it would be very interesting to identify some other perspectives (e.g. social or environmental points of view).
[Authors answer 5]. We have incorporated new paragraphs in section 4 to improve the main findings from multiples angles: from local/social perspectives (lines 1063-1067); form environmental perspective (lines1067-1069 and 1073-1077); and from individual/social perspectives (lines 1070-1073). The main findings from the government perspective can be found on section 4.3 (lines 1078-1091). Likewise, other main findings from environmental and social perspectives can be found on section 4.2 (lines 1000-1017) and 4.3 (1078-1087). In addition we have renamed the 4.1 section to avoid confusion about where to find main research findings throughout the discussion section 4.
